# AI2TALE: An Innovative Information Theory-based Approach for Learning to Localize Phishing Attacks

**Van Nguyen**[1,2]  **Tingmin Wu**[2]  **Xingliang Yuan**[3]  **Marthie Grobler**[2]
**Surya Nepal**[2]  **Carsten Rudolph**[1]
[1]Monash University  [2]CSIRO's Data61  [3]The University of Melbourne, Australia

## Abstract

Phishing attacks remain a significant challenge for detection, explanation, and defense, despite over a decade of research on both technical and non-technical solutions. AI-based phishing detection methods are among the most effective approaches for defeating phishing attacks, providing predictions on the vulnerability label (i.e., phishing or benign) of data. However, they often lack intrinsic explainability, failing to identify the specific information that triggers the classification. To this end, we propose AI2TALE, an innovative deep learning-based approach for email (the most common phishing medium) phishing attack localization. Our method aims to not only predict the vulnerability label of the email data but also provide the capability to automatically learn and identify the most important and phishing-relevant information (i.e., sentences) in the phishing email data, offering useful and concise explanations for the identified vulnerability.

Extensive experiments on seven diverse real-world email datasets demonstrate the capability and effectiveness of our method in selecting crucial information, enabling accurate detection and offering useful and concise explanations (via the most important and phishing-relevant information triggering the classification) for the vulnerability of phishing emails. Notably, our approach outperforms state-of-the-art baselines by 1.5% to 3.5% on average in Label-Accuracy and Cognitive-True-Positive metrics under a weakly supervised setting, where only vulnerability labels are used without requiring ground truth phishing information.

## 1 Introduction

Phishing attacks (i.e., attempts to deceitfully get personal and financial information such as usernames, passwords, and bank accounts through electronic communication with malicious intentions) have become a serious issue. Nowadays, there are various ways to conduct phishing attacks, while the most common method is through the use of emails. Email phishing is crafted to trigger psychological reactions in the users by using persuasion techniques via cognitive principles (Van Der Heijden & Allodi, 2019) such as scarcity, consistency, and authority. According to recent reports (Federal Bureau of Investigation, 2022; Anti-Phishing Working Group, 2022), there has been a notable rise in the occurrence of increasingly sophisticated phishing attacks, presenting more formidable challenges for detection and defense.

The widespread adoption of AI (i.e., using machine learning and deep learning approaches) has brought substantial influences and great success in various domain applications such as autonomous driving (Chen et al., 2023a), data generation (Devlin et al., 2019; Radford et al., 2019; Raffel et al., 2020), drug discovery (Paul et al., 2021), and software vulnerability and malware detection (Li et al., 2018; Nguyen et al., 2022; 2024; Bensaoud et al., 2024). By leveraging the power of machine learning and deep learning, there have been many efforts proposed for solving phishing attack problems from the investigation of the cognitive bias's impact used in email phishing (Parsons et al., 2015; Van Der Heijden & Allodi, 2019) to phishing attack detection (Ramanathan & Wechsler, 2013; Tyagi et al., 2018; Rao & Pais, 2019; Yang et al., 2019; Xiao et al., 2020; Lin et al., 2021; Asiri et al., 2024). Recent advancements in natural language processing (NLP) and the growing popularity of large language models (LLMs), such as BERT (Devlin et al., 2019), GPT-2 (Radford et al., 2019),

T5 (Raffel et al., 2020), and GPT-4 (OpenAI et al., 2024), have led to initial attempts to leverage these models for phishing attack mitigation. For instance, scam-baiting mail servers were developed to autonomously engage in scam-baiting activities (Chen et al., 2023b). Additionally, approaches for devising and detecting phishing emails using LLMs were introduced in (Heiding et al., 2024).

It has been proven that utilizing AI-based approaches (i.e., machine learning and deep learning-based algorithms) to detect phishing attacks in the early stages is one of the effective solutions for preventing and reducing the negative effects caused (Basit et al., 2020; Naqvi et al., 2023; Asiri et al., 2024). Although AI-based phishing detection methods can predict the vulnerability label of the data (i.e., phishing or benign) and have shown promising performances, they often lack the intrinsic ability to provide explanations that offer concise and meaningful interpretations (i.e., the information causing the data phishing) to the users. Motivated by this problem, in this paper, we study the following topical research question:

> In addition to predicting the vulnerability label (i.e., phishing or benign) of the data, how to derive an effective deep learning-based method that also has the capability to automatically learn and identify the most important and phishing-relevant information (i.e., sentences) triggering the classification for providing a useful and concise explanation about the vulnerability of the phishing data to users? (in our paper, we name this problem as phishing attack localization)

To this end, in the scope of our paper, we study the phishing attack localization problem on emails (i.e., the most common medium for phishing attacks) where we propose an innovative information theory-based model to solve the problem. Our method can detect the vulnerability (i.e., phishing or benign) of the email data as well as automatically learn and identify the most important and phishing-relevant information (e.g., sentences) in phishing emails. The selected information helps provide useful and concise explanations about the vulnerability of the phishing data. It is worth noting that the ability to identify the important and phishing-relevant information that causes the email data to be classified as phishing to provide a corresponding useful and concise interpretation is the main difference between phishing attack localization and phishing attack detection.

In summary, our key contributions are as follows:

- We study an important problem of phishing attack localization aiming to tackle and improve the explainability (transparency) of email phishing detection. Automated machine learning and deep learning-based techniques for this problem have not yet been well studied.
- We propose AI2TALE, an innovative deep learning-based framework derived from an information-theoretic perspective and information bottle-neck theory for phishing attack localization. Our method can work effectively in a weakly supervised setting (details in Section 3.1), hence providing an important practical solution for defeating phishing attacks.
- Based on the explainable machine learning and email phishing domain knowledge, we propose using appropriate measures, including *Label-Accuracy* and *Cognitive-True-Positive* (please refer to Section 4.2 for details), for the phishing attack localization problem.
- We comprehensively evaluate our method on seven real-world diverse email datasets. The rigorous and extensive experiments demonstrate the effectiveness and superiority of our method over the state-of-the-art baselines.

## 2 RELATED WORK

Phishing detection methods (Le et al., 2018; Sahingoz et al., 2019; Li et al., 2019; Das et al., 2019; Abdelnabi et al., 2020; Zamir et al., 2020; Alam et al., 2020; Yang et al., 2021; Salahdine et al., 2021; Lin et al., 2021; Chrysanthou et al., 2024; Asiri et al., 2024; Heiding et al., 2024) have been widely applied to detect phishing attacks, helping to prevent and mitigate their negative effects. While achieving promising performances, they often lack the intrinsic ability to provide concise and meaningful explanations (i.e., the information causing the data phishing) to the users.

Automated deep learning-based techniques for the phishing attack localization problem have not yet been well studied. The interpretable machine-learning research appears to be an appropriate direction for addressing the phishing attack localization problem. In short, interpreting approaches (e.g., (Caruana et al., 2015; Rich, 2016; Marco T. Ribeiro, 2016; Chen et al., 2018; Bang et al., 2021; Yoon et al., 2019; Nguyen et al., 2021; Jethani et al., 2021; Vo et al., 2023a; Qian et al., 2024; Choi et al., 2024)) are used for explaining the behavior of deep learning-based systems or

the ground truth label of the data by automatically learning and identifying the most important information (e.g., attributions or features) existing from the data that are responsible in causing the corresponding decision of black-box models and the ground truth label. For example, in sentiment analysis, interpretable machine-learning approaches (e.g., (Chen et al., 2018; Vo et al., 2023b)) help to give a comprehensive explanation for a movie review (positive or negative) by identifying and highlighting the most important keywords or sentences.

In practice, explaining models can be divided into two categories including "post-hoc explainability techniques" and "intrinsic explainability techniques". Post-hoc explainability techniques (e.g., LIME (Ribeiro et al., 2016) and SHAP (Lundberg & Lee, 2017)) aim to elucidate the decisions of a black-box model (e.g., a deep learning model, where the internal workings are not easily understandable or interpretable) without modifying the model itself. The techniques are often applied externally to the black-box model to generate explanations specific to its predictions but do not offer a comprehensive understanding of the black-box model's internal architecture. In contrast, intrinsic explainability techniques, a.k.a. self-explanatory models, (e.g., deep neural network-based methods with interpretable components such as L2X (Chen et al., 2018) and AIM (Vo et al., 2023b)) are integrated directly into a model architecture, providing interpretability as inherent features, and offering explainability as part of their design.

It is evident that intrinsic interpretable machine learning methods, a.k.a. self-explanatory models, are strongly suitable for phishing attack localization because they are not only able to make predictions themselves but also to automatically learn and identify the most important information of the data obtained from the models to explain the model's prediction decision. In our paper, we compared the performance of our method with several recent, popular, and state-of-the-art interpretable machine learning approaches falling into the category of intrinsic interpretable models including L2X (Chen et al., 2018), INVASE (Yoon et al., 2019), ICVH (Nguyen et al., 2021), VIBI (Bang et al., 2021), and AIM (Vo et al., 2023b) (refer to the appendix, Section 6.4, for their brief descriptions).

## 3 THE PROPOSED APPROACH

### 3.1 THE PROBLEM STATEMENT

We denote an email data sample as $X = [\mathbf{x}_1, ... \mathbf{x}_L]$ consisting of $L$ sentences. In the scope of our paper, we consider each email as a sequence of sentences. We assume that $X'$s vulnerability label $Y \in \{0, 1\}$ (i.e., 1: phishing and 0: benign). In the context of phishing attack localization, we aim to develop an automatic AI-based approach that can not only detect the vulnerability $Y$ of the email $X$ but also provide the capability to automatically learn and identify the important and phishing-relevant information (i.e., sentences) denoted by $\tilde{X}$ (a subset of $X$) causing $X$ phishing.

It is worth noting that for almost all publicly available phishing-relevant data (e.g., emails), there are only labels related to the data's vulnerability (phishing or benign) by domain experts with the assistance of machine learning or deep learning tools. We almost do not have the ground truth of phishing information (i.e., the information truly causes the data to be classified as phishing). *In the phishing attack localization problem, we name this context as a weakly supervised setting* where during the training process, we only use the vulnerability label $Y$ of the data while not requiring the ground truth of phishing information of the data.

### 3.2 METHODOLOGY

Here we present how our AI2TALE method, named after the initials of some words in the paper's title, works and addresses the phishing attack localization problem to tackle and improve the explainability (transparency) of phishing detection. An overall visualization is depicted in Figure 1.

#### 3.2.1 LEARNING TO SELECT THE IMPORTANT AND PHISHING-RELEVANT INFORMATION AND THE TRAINING PRINCIPLE

**Phishing-relevant information selection process**  As shown in Figure 1, the first part of our method is the selection network $\zeta$. It aims to learn and identify the most important and label-relevant information (i.e., sentences) in each email in an automatic and trainable manner. Note that, in terms of the phishing email, the key selected information stands for the phishing-relevant information causing the email phishing.

Given an email $X$ consisting of $L$ sentences from $\mathbf{x}_1$ to $\mathbf{x}_L$ (i.e., *each sentence $\mathbf{x}_i$ is represented as a vector using a learnable embedding method, refer to the data processing and embedding in the appendix, Section 6.2, for details*), to identify the important and label-relevant sentences $\tilde{X}$ in $X$, we introduce a selection process $\zeta$ (i.e., *it is learnable and maps $\mathbf{R}^L \mapsto [0,1]^L$*) aiming to learn a set of independent Bernoulli latent variables $\mathbf{z} \in \{0,1\}^L$ representing the importance of the sentences to the email's vulnerability $Y$. Specifically, each element $z_i$ in $\mathbf{z} = \{z_i\}_{i=1}^L$ indicates whether $\mathbf{x}_i$ is related to the vulnerability $Y$ of $X$ (i.e., if $z_i$ is equal to 1, the sentence $\mathbf{x}_i$ plays an important role).

We model $\mathbf{z} \sim \mathrm{MultiBernoulli}(\mathbf{p}) = \prod_{i=1}^L \mathrm{Bernoulli}(p_i)$, indicating $\boldsymbol{x}_i$ is related to the vulnerability $Y$ with probability $p_i$, where $p_i = \omega_i(X;\alpha)$. Here, $\omega$ is a neural network parameterized by $\alpha$ (i.e., $\omega$ takes $X$ as input and outputs corresponding $\mathbf{p} = \{p_i\}_{i=1}^L$). With $\mathbf{z}$, we construct $\tilde{X} = \zeta(X)$ (i.e., the subset statements that lead to the vulnerability $Y$) by $\tilde{X} = X \odot \mathbf{z}$, where $\odot$ represents the element-wise product. To make this computational process (involving sampling from a Multi-Bernoulli distribution) continuous and differentiable during training, we apply the Gumbel-Softmax trick (Jang et al., 2016; Maddison et al., 2016) for relaxing each Bernoulli variable $z_i$ (refer to the appendix, Section 6.1, for de-

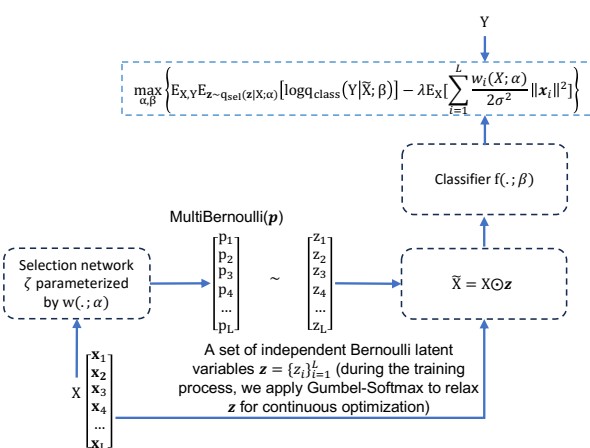

Figure 1: Visualization of our AI2TALE method.

tails). Note that we can view the selector model as a distribution $q_{sel}(\mathbf{z}|X;\alpha)$ over a selector variable $\mathbf{z}$, which indicates the important features for a given sample $X$.

**Mutual information for guiding the selection process** In information theory (Shannon, 1998; Cover & Thomas, 2006), mutual information is used to measure the mutual dependence between two random variables. In particular, it quantifies the information obtained about one random variable by observing the other. To illustrate, consider a scenario where A denotes the outcome of rolling a standard 6-sided die, and B represents whether the roll results in an even number (0 for even, 1 for odd). Evidently, the information conveyed by B provides insights into the value of A, and vice versa. In other words, these random variables possess mutual information.

Leveraging this property of mutual information and inspired by (Chen et al., 2018; Nguyen et al., 2021), we maximize the mutual information between $\tilde{X}$ and $Y$ as mentioned in Eq. (1) with the intuition is that by using the information from $Y$, the selection process $\zeta$ will be learned and enforced to obtain the most meaningful $\tilde{X}$ (i.e., $\tilde{X}$ can predict the vulnerability $Y$ of $X$ correctly). By viewing $\tilde{X}$ and $Y$ as random variables, the selection process (model) $\zeta$ is learned by maximizing the mutual information between $\tilde{X}$ and $Y$ as follows:

$$\max_{\zeta} \mathbf{I}(\tilde{X}, Y) \tag{1}$$

Following (Cover & Thomas, 2006), we expand Eq. (1) as the Kullback-Leibler divergence, measuring the relative entropy or difference in information represented by two distributions, of the product of marginal distributions of $\tilde{X}$ and $Y$ from their joint distribution:

$$\mathbf{I}(\tilde{X}, Y) = \int p(\tilde{X}, Y) \log \frac{p(\tilde{X}, Y)}{p(\tilde{X})p(Y)} d\tilde{X} dY \geq \int p(Y, \tilde{X}) \log \frac{q(Y \mid \tilde{X})}{p(Y)} dY d\tilde{X}$$

In practice, estimating mutual information is challenging as we typically only have access to samples but not the underlying distributions. Therefore, in the above derivation, we apply to use a variational distribution $q(Y|\tilde{X})$ to approximate the posterior $p(Y|\tilde{X})$, hence deriving a variational lower bound of $\mathbf{I}(\tilde{X}, Y)$ for which the equality holds if $q(Y|\tilde{X}) = p(Y|\tilde{X})$. This can be further expanded as:

$$\mathbf{I}(\tilde{X}, Y) \geq \int p(Y, \tilde{X}, X) \log \frac{q(Y \mid \tilde{X})}{p(Y)} dY d\tilde{X} dX = \mathbf{E}_{X,Y} \mathbf{E}_{\tilde{X}|X} [\log q(Y|\tilde{X})] + \text{const} \quad (2)$$

To model the conditional variational distribution $q(Y|\tilde{X})$, we introduce a classifier implemented with a neural network $f(\tilde{X}; \beta)$, which takes $\tilde{X}$ as input and outputs its corresponding label $Y$ (i.e., we view the classifier model as a distribution $q_{class}(Y|\tilde{X}; \beta)$). Our objective is to learn the selection process (model) as well as the classifier to maximize the following objective function:

$$\max_{\alpha, \beta} \left\{ \mathbf{E}_{X,Y} \mathbf{E}_{\mathbf{z} \sim q_{sel}(\mathbf{z}|X; \alpha)} [\log q_{class}(Y|X \odot \mathbf{z}; \beta)] \right\} \quad (3)$$

The mutual information facilitates a joint training process for the classifier and the selection process. The classifier learns to identify a subset of features leading to a data's label while the selection process is designed to select the best subset of features according to the feedback of the classifier.

### 3.2.2 BENEFITS AS WELL AS POTENTIAL WEAKNESSES OF THE MUTUAL INFORMATION TRAINING PRINCIPLE AND OUR INNOVATIVE SOLUTIONS

With the training principle mentioned in Eq. (1), we aim to maximize the mutual information between $\tilde{X}$ and $Y$ for guiding the whole training process to identify the sentences related to an email's vulnerability. Recall that during this training process, the classifier learns to identify a subset of sentences leading to an email sample's vulnerability label while the selection process is designed to select the best subset according to the feedback of the classifier.

*In short, this joint training process between the classifier $f(., \beta)$ and the selection network $\omega(., \alpha)$ brings benefits for selecting the important and phishing-relevant sentences from phishing emails; However, we observe two potential limitations of this training principle as follows:*

**Obtaining a superset of phishing-relevant sentences** *The first limitation of the training principle mentioned in Eq. (1) is that it does not theoretically guarantee to eliminate sentences unrelated to the vulnerability of a specific email.* Therefore, the set of selected sentences can be a superset of the true phishing-relevant sentences. In the worst case, a selection process can always select all the sentences in an email, which is still a valid solution for the above maximization mentioned in Eq. (1). To deal with this problem and enable the model to successfully select and highlight truly important and phishing-relevant information, inspired by (Tishby et al., 2000; Slonim & Tishby, 2000) about using information bottleneck theory, we propose an additional term for training the selection process (model) $\zeta$, derived from the following principle:

$$\max_{\zeta} (\mathbf{I}(\tilde{X}, Y) - \lambda \mathbf{I}(X, \tilde{X})) \quad (4)$$

where $\lambda$ is a hyper-parameter indicating the weight of the second mutual information.

By minimizing the mutual information between $X$ and $\tilde{X}$, we encourage $\tilde{X}$ to be as "different" to $X$ as possible. In other words, the selection process prefers to select a smaller subset that excludes the sentences unrelated to vulnerability $Y$ of the corresponding data $X$. Accordingly, we can derive an upper bound of the minimization of the mutual information between $X$ and $\tilde{X}$:

$$\mathbf{I}(\tilde{X}, X) = \int p(\tilde{X}, X) \log \frac{p(\tilde{X}|X)}{p(\tilde{X})} d\tilde{X} dX \leq \mathbf{E}_X \mathbf{E}_{\tilde{X}|X} [\log \frac{p(\tilde{X}|X)}{r(\tilde{X})}] \quad (5)$$

for any distribution $r(\tilde{X})$.

We then further derive $\mathbf{I}(\tilde{X}, X)$ as the Kullback-Leibler divergence of the product of marginal distributions of $\tilde{X}$ and $X$ from their joint distribution:

$$\mathbf{E}_X \mathbf{E}_{\tilde{X}|X} [\log \frac{p(\tilde{X}|X)}{r(\tilde{X})}] = \mathbf{E}_X \mathbf{E}_{\tilde{X}|X} [\sum_{i=1}^{L} \log \frac{p(\tilde{\boldsymbol{x}}_i|X)}{r(\tilde{\boldsymbol{x}}_i)}] = \sum_{i=1}^{L} \mathbf{E}_X [D_{KL}(p(\tilde{\boldsymbol{x}}_i|X) \| r(\tilde{\boldsymbol{x}}_i))] \quad (6)$$

Minimizing $\mathbf{I}(\tilde{X}, X)$ is now equivalent to minimizing the KL divergence between $p(\tilde{\boldsymbol{x}}_i|X)$ and $r(\tilde{\boldsymbol{x}}_i)$ with $i$ from 1 to $L$. Therefore, one can view $r(\tilde{\boldsymbol{x}}_i)$ as the prior distribution, which is constructed by $r(\tilde{\boldsymbol{x}}_i) = \mathcal{N}(\tilde{\boldsymbol{x}}_i|0, \sigma^2)$. Given that $p(\tilde{\boldsymbol{x}}_i|X)$ is a Gaussian mixture distribution (i.e., between $p_i\mathcal{N}(\tilde{\boldsymbol{x}}_i|\boldsymbol{x}_i, \sigma^2)$ and $(1 - p_i)\mathcal{N}(\tilde{\boldsymbol{x}}_i|0, \sigma^2)$ where $\sigma > 0$ is a small number), the intuition is that the prior prefers the small values centered at 0. In this way, $p(\tilde{\boldsymbol{x}}_i|X)$ is encouraged to select fewer sentences. $D_{KL}(p(\tilde{\boldsymbol{x}}_i|X)\|r(\tilde{\boldsymbol{x}}_i))$ can be computed by the following approximation (Gal & Ghahramani, 2016):

$$\frac{\omega_i(X; \alpha)}{2\sigma^2} \|\boldsymbol{x}_i\|^2 + (\log \sigma + \frac{1}{2}\sigma^2) + \text{const} \tag{7}$$

Combining $\mathbf{I}(\tilde{X}, Y)$ and $\mathbf{I}(X, \tilde{X})$ by $\max\left(\mathbf{I}(\tilde{X}, Y) - \lambda\mathbf{I}(X, \tilde{X})\right)$, we get a unified training objective:

$$\max_{\alpha, \beta}\{\mathbf{E}_{X,Y}\mathbf{E}_{\mathbf{z}\sim q_{sel}(\mathbf{z}|X;\alpha)}[\log q_{class}(Y|X \odot \mathbf{z}; \beta)] - \lambda\mathbf{E}_X[\sum_{i=1}^{L}\frac{\omega_i(X; \alpha)}{2\sigma^2}\|\boldsymbol{x}_i\|^2]\} \tag{8}$$

**Encoding the vulnerability label via its selections instead of via truly meaningful information** The selected features obtained from the joint training process of the classifier $f(., \beta)$ and the selection network $\omega(., \alpha)$ cause the other potential limitation for the training principle mentioned in Eq. (1). In particular, the predictions of the classifier $f(., \beta)$ can be based more on the features selected from the selection network $\omega(., \alpha)$ than the underlying information contained in the features. In this case, the selected information (i.e., sentences) can be any subset of the sentences and may be less likely to be meaningful subsets from the data.

To deal with this problem and ensure the learnable selection process respecting the data distribution to select the meaningful and label-relevant information (e.g., sentences) of the data, in addition to learning the classifier jointly with the selection network as mentioned in Eq. (8), inspired by (Jethani et al., 2021), we propose to learn the classifier model $f(., \beta)$ disjointly to approximate the ground truth conditional distribution of $Y$ given $X_R$ where $X_R = X \odot \mathbf{r}$ with $\mathbf{r} \sim \text{MultiBernoulli}(0.5)$ (denoted by $\mathbf{r} \sim \text{B}(0.5)$ for short). *This procedure helps adjust the classifier to let it not only be affected by the information obtained from the selection network but also based on the information from the data to update its parameters*. That helps prevent the problem of encoding the vulnerability label via its selections to improve the data representation learning process. We name this procedure as a data-distribution mechanism. This procedure is expressed as learning $q_{class}(.; \beta)$ to maximize:

$$\mathbf{E}_{X,Y}\mathbf{E}_{\mathbf{r}\sim\text{B}(0.5)}\{\log q_{class}(Y|X \odot \mathbf{r}; \beta)\} \tag{9}$$

To make this computational procedure (i.e., it consists of sampling operations from a Multi-Bernoulli distribution) continuous and differentiable during the training process, we apply the temperature-dependent Gumbel-Softmax trick (Jang et al., 2016; Maddison et al., 2016) for relaxing each Bernoulli variable $r_i \in \mathbf{r}$ using the RelaxedBernoulli distribution function from TensorFlowAPI (2023). Noting that in Eq.(9), by setting $\mathbf{r} \sim \text{B}(0.5)$ to randomly select sentences from each email instead of using the entire email content, we also aim to introduce randomness into the training corpus, akin to data augmentation, which helps enhance the generalization capability of the selection network that facilitates the performance of the classifier.

### 3.2.3 A SUMMARY OF OUR AI2TALE METHOD

Algorithm 1 shows the details of our proposed AI2TALE method in the training phase. *It is worth noting that during the training process, our model is trained to learn and identify the most important and phishing-relevant sentences in corresponding emails without using any information about the ground truth of phishing-relevant sentences*. This shows a great advantage of using our method for phishing attack localization in real-world scenarios because in practice, to almost all publicly available email datasets, there is only information about the vulnerability $Y$ (i.e., phishing or benign) of the data by domain experts with the help of machine learning and deep learning tools.

**The inference (testing) phase** After the training phase, the selection network $\zeta$ is capable of selecting the most important and phishing-relevant sentences of a given email data $X$ by offering a high value for the corresponding coordinates $\omega_i(X; \alpha)$, meaning that $\omega_i(X; \alpha)$ represents the

influence level of the sentence $\boldsymbol{x}_i$. We hence can pick out the most relevant sentences based on the magnitude of $\omega_i(X; \alpha)$. Using the selected information, the trained classifier then can predict the vulnerability of the associated email data. *In our paper, with the aim of providing the most highly qualified and concise explanation of the vulnerability of email data to users, **we primarily assess the model's performance based on the most important (top-1) selected sentence from each email.***

---

**Algorithm 1:** The algorithm of our proposed AI2TALE method for the phishing attack localization problem.

---

**Input:** An email dataset $S = \{(X_1, Y_1), \ldots, (X_{N_S}, Y_{N_S})\}$ where each email $X_i = [\mathbf{x}_1, \ldots \mathbf{x}_L]$ consisting of $L$ sentences while its label $Y_i \in \{0, 1\}$ (i.e., 1: phishing and 0: benign). We denote the number of training iterations $nt$; the mini-batch size $m$; the trade-off hyper-parameter $\lambda$. We randomly partition $S$ into three different sets including the training set $S_{train}$ (for training the model), the validation set $D_{val}$ (for model selection during training), and the testing set $D_{test}$ (for evaluating the model).

1 We initialize the parameters $\alpha$ and $\beta$ of the selection model $\zeta$ (i.e., parameterized by $\omega(., \alpha)$) and the classifier model $f(., \beta)$, respectively.

2 **for** $t = 1$ *to* $nt$ **do**

3      Choose a mini-batch of embedded emails denoted by $\{(X_i, Y_i)\}_{i=1}^{m}$.

4      Update the classifier's parameter $\beta$ via minimizing the following cross-entropy loss $\mathbb{E}_{X,Y}\mathbb{E}_{\mathbf{r} \sim B(0.5)}[\mathcal{L}_{ce}(Y, f_\beta(X \odot \mathbf{r})]$ using Adam optimizer (Kingma & Ba, 2014). Minimizing this function is equivalent to maximizing the objective function in Eq. (9).

5      Update the classifier's parameter $\beta$ and the selection model parameter's $\alpha$ via minimizing the following objective function $\mathbb{E}_{X,Y}\mathbb{E}_{\mathbf{z} \sim q_{sel}(\mathbf{z}|X;\alpha)}[\mathcal{L}_{ce}(Y, f_\beta(X \odot \mathbf{z}))] + \lambda\mathbb{E}_X[\sum_{i=1}^{L} \frac{\omega_i(X;\alpha)}{2\sigma^2}\|\boldsymbol{x}_i\|^2]$ using Adam optimizer. Minimizing this function is equivalent to maximizing the objective function in Eq. (8).

6 **end**

**Output:** The trained model for phishing attack localization.

---

# 4 EXPERIMENTS

## 4.1 STUDIED DATASETS

We conducted experiments on seven diverse real-world email datasets including **IWSPA-AP** (i.e., the dataset was collected as part of a shared task to address phishing scam emails), **Nazario Phishing Corpus** (i.e., the dataset was received by one user, comprising a diverse collection of phishing emails), **Miller Smiles Phishing Email** (i.e., the dataset contains various examples of phishing emails to trick recipients into engaging with malicious links or providing sensitive information), **Phish Bowl Cornell University** (i.e., the dataset contains phishing emails that have been spotted and reported by students and staff at Cornell University), **Fraud emails** (i.e., the dataset contains fraudulent emails attempting Nigerian Letter where all the emails are in one text file and contain a large amount of header data), **Cambridge** (i.e., the dataset contains a large number of email headers (involving information such as sending and receiving addresses and email subjects) and the body content of phishing emails), and **Enron Emails** (i.e., the dataset was released as part of an investigation into Enron, consisting of emails from mostly senior management of Enron). Refer to the appendix, Section 6.2, for details on the links, processing, and characteristics of these datasets.

## 4.2 MEASURES

In addition to our AI2TALE method, interpretable machine learning approaches, particularly intrinsic interpretable models, can be adopted and applied to solve the email phishing attack localization problem. Our study also introduces and discusses appropriate measures to evaluate the performance of our method and the baselines, which is another contribution of our work.

To evaluate the performance of our AI2TALE method and the baselines in phishing attack localization, based on the explainable machine learning and email phishing domain knowledge, we introduce and utilize two main metrics including **Label-Accuracy** and **Cognitive-True-Positive**.

Via the **Label-Accuracy** metric, we measure whether the selected sentences obtained from each model are truly important and help accurately predict the true vulnerability label (i.e., phishing or benign) of the associated emails. *The intuition is that the most important sentences contribute the most to the emails' vulnerability, especially to phishing emails.* In our experiments, we assess each model's top-1 selected sentence and measure if this sentence can effectively predict the email's vulnerability without considering all sentences in the email. The higher the value of the Label-Accuracy measure, the better the model's performance in identifying and selecting crucial and label-relevant information from the data. For the **Cognitive-True-Positive** metric, we investigate if the top-1 selected sentences from each method also reflect the human cognitive principles, exploiting psychological triggers to deceive recipients, used in phishing emails (i.e., Reciprocity, Consistency, Social Proof, Authority, Liking, and Scarcity (Akbar, 2014; Butavicius et al., 2015; Ferreira et al., 2015; Heijden & Allodi, 2019) based on the associated keywords, *refer to the appendix, Section 6.3, for the computation*). In our experiments, we consider the most important (top-1) selected sentences of phishing emails and calculate how many percent of these reflect and consist of cognitive triggers. The source code and data for reproducing the experiments of our AI2TALE method are publicly available at `https://github.com/vannguyennd/ai2tale-iclr`.

## 4.3 BASELINE METHODS

The baselines of our method are recent, popular, and state-of-the-art interpretable machine learning approaches falling into the category of intrinsic interpretable models including L2X (Chen et al., 2018), INVASE (Yoon et al., 2019), ICVH (Nguyen et al., 2021), VIBI (Bang et al., 2021), and AIM (Vo et al., 2023b) that we apply to solve the email phishing attack localization problem. As mentioned in Section 2, intrinsic interpretable machine learning techniques, a.k.a. self-explanatory models, are strongly suitable for phishing attack localization because they can not only make predictions themselves but also identify the most important features of the data to explain the model's predictive decision. See the appendix, Section 6.4, for the baselines' descriptions.

## 4.4 EXPERIMENTAL RESULTS

We compare the performance of our AI2TALE method with the baselines including **L2X** (Chen et al., 2018), **INVASE** (Yoon et al., 2019), **ICVH** (Nguyen et al., 2021), **VIBI** (Bang et al., 2021), and **AIM** (Vo et al., 2023b) in the task of phishing attack localization (in terms of not only predicting the vulnerability of the email data but also learning and identifying the most important (top-1) and phishing-relevant information existing in the data. The selected information helps provide useful and concise explanations about the vulnerability of the phishing email data for the users) using the Label-Accuracy and Cognitive-True-Positive measures.

**Quantitative Results**   The experimental results in Table 1 show that our AI2TALE method obtains the best performance on both Label-Accuracy and Cognitive-True-Positive compared to the baselines. Importantly, our method achieves a significantly higher performance, with improvements ranging from approximately 1.5% to 3.5% compared to state-of-the-art baselines, measured by the combined average performance of two main metrics, i.e., Label-Accuracy and Cognitive-True-Positive. The results demonstrate the effectiveness and advancement of our method for phishing attack localization in learning and identifying the most meaningful and crucial information leading to the vulnerability of the email data, especially for phishing ones, compared to the baselines.

Table 1: The performance of our AI2TALE method and the baselines for the Label-Accuracy (Label-Acc) and Cognitive-True-Positive (Cognitive-TP) measures, as well as their combined average results (denoted as Average), on the testing set (the best results in **bold**).

| Methods | Label-Acc | Cognitive-TP | Average |
|---|---|---|---|
| INVASE (Yoon et al., 2019) | 98.30% | 97.20% | 97.75% |
| ICVH (Nguyen et al., 2021) | 96.72% | 98.10% | 97.41% |
| L2X (Chen et al., 2018) | 98.25% | 97.20% | 97.73% |
| VIBI (Bang et al., 2021) | 96.65% | 94.99% | 95.82% |
| AIM (Vo et al., 2023b) | 98.40% | 97.10% | 97.75% |
| AI2TALE (Ours) | **99.33%** | **98.95%** | **99.14%** $\uparrow \sim (1.5\% \rightarrow 3.5\%)$ |

In addition, the results on the Cognitive-True-Positive measure of our AI2TALE method and the baselines shown in Table 1 also indicate that the most important (top-1) selected sentences from our method and the baselines also reflect the cognitive triggers used in phishing emails. Compared to the baselines, our proposed AI2TALE method achieves the highest performance on the Cognitive-True-Positive measure. Under the weakly supervised setting, when dealing with complex data (e.g., emails written from various writing styles, structures, and sources) and only the most important (top-1) selected sentences from emails are utilized, the observed improvement of our AI2TALE method from around 1.5% to 3.5% in the combined average performance of the Label-Accuracy and Cognitive-True-Positive measures, especially within the range over 99% and approaching 100%, signifies a substantial advancement.

Table 2: The ground truth and the predicted label based on the most important (top-1) selected sentence (highlighted in yellow) of each email from our AI2TALE method. The top-1 selected sentence provides concise and useful information. However, to further aid in label explanation, we also show the second and third most relevant sentences. See the appendix, Section 6.7, for details.

| Truth | Model | Emails |
|---|---|---|
| phishing | phishing | **The email:** "Please confirm your online banking records! Dear customer of natwest bank, we are running a scheduled maintenance on our servers. We want to make sure your money and your personal details are safe and secure. Due to new security policies all natwest bank customers must complete the natwest customer form. To complete the form, please use the link below natwest customer form this should take you directly to the natwest customer form. Sincerely, natwest customer service good." 
 **Note:** The phishing email poses as a legitimate message from natwest bank, requesting confirmation of banking records. Its goal is to trick individuals into providing personal information under the guise of security, emphasizing the need to verify the authenticity of such communications. Our method successfully figures out the sentence "Please confirm your online banking records!" representing the key message of the phishing email. It serves to create urgency and prompt the recipient to take action, suggesting that there is a need to verify their record information. |
| phishing | phishing | **The email:** "Temporarily suspended. Dear customer, customer advice, please address the following issues. The details that you have entered have not been recognized. For your security, your online service has been temporarily locked. No further attempts will be accepted. If you provide us with the following details, you should be able to access the service in just a few minutes. Click here to get started legal info privacy security 2005 2010." 
 **Note:** This phishing email tries to convey (i) the action taken is intended to protect the recipient's well-being and (ii) a sense of urgency, encouraging the recipient to address the issue promptly to regain access to their online service. Our method successfully figures out the sentence "For your security, your online service has been temporarily locked." representing the key message of the phishing email. |
| phishing | phishing | **The email:** "Bulk attention! Your discover account will close soon! Dear member, we have faced some problems with your account, so please update the account. If you do not update will be closed. To update your account, just confirm your information. (it only takes a minute). It's easy. 1. Click the link below to open a secure browser window. 2. Confirm that you're the owner of the account, and then follow the instructions." 
 **Note:** The message from the selected sentence obtained from our method exhibits cognitive triggers commonly associated with phishing attempts used in the phishing email. In particular, it implies a sense of urgency (concern) via problems with your account while "Dear member" aims to establish a connection with the recipient and imply that the message comes from a trusted source. The phrase "please update the account" creates a sense of familiarity and consistency. |
| benign | phishing | **The email:** "Lower your mortgage payment! Bad credit, no problem! Lower your mortgage payment! Bad credit, no problem! Whether your credit is excellent or less than perfect, loanweb has a lender that can help you! Lowest rates on the web! Bad credit? Refinance to get cash! Lower monthly payments! Bad credit? Refinance to consolidate bills! Click here for lower mortgage payment! Win a free bread maker! Win free bread makers, toaster ovens, cookware and more get a free subscription to cooking pleasures magazine get a free multi purpose grater test and keep free cooking products get free recipes from world famous chefsplus get a free 90 day membership in the cooking club of america. Home equity loans without perfect credit! Click here for a home equity loan! Free application! Home equity loans up to 125 potential tax deductible interest! Customized, competitive equity lines and loans. Apply now!" 
 **Note:** In the datasets we utilized, the ground truth label for this email is benign. However, our method predicts it as phishing due to the detection of potential phishing indicators, such as the use of enticing offers of "lowering mortgage payments" and the presence of "suspicious links" potentially prompting users to provide personal information and bank account details via the phrase "Click here for lower mortgage payment!". Although the email's ground truth classification was incorrect, we deem the predicted label and the highlighted sentence from our model valuable in alerting users to potential phishing attempts. |

**Qualitative Results**   To further demonstrate the effectiveness of our AI2TALE method in solving the phishing attack localization problem, in Table 2, we present various email samples alongside the most important (top-1) selected sentences extracted by our method as well as the predicted labels of the corresponding emails based on the selected sentences. Via these qualitative results, our method showcases *its effectiveness in learning and identifying the most important and phishing-relevant information (i.e., sentences) from phishing emails.* That helps provide useful and concise interpretations for the phishing prediction, offering valuable insights into the nature of the attacks.

In addition, to investigate the characteristics of the selected information in false positive examples (i.e., where our method incorrectly predicts emails as phishing when they are benign, resulting in a false positive rate of 0.451% and a false negative rate of 0.899%), we present a representative example in Table 2. Our analysis reveals that the model identified potential phishing elements within the email, despite the misclassification. Notably, the predicted label and sentence still offer valuable insights, enabling users to take proactive measures to enhance security.

**Human evaluation**  We conduct a human evaluation to investigate the usefulness of our AI2TALE  method in identifying the most important and phishing-related information (e.g., sentences) in phishing emails to help provide useful and concise explanations about the phishing of the corresponding email data. In particular, we evaluate whether the most important (top-1) selected sentence in each phishing email by our proposed AI2TALE method is perceived as convincing information for email users. To do that, we asked participants to evaluate the selected sentences of 10 different phishing emails (randomly chosen from the testing set) in terms of whether the sentence selected in each email is important to influence and persuade users to follow the instructions in the emails (refer to the appendix, Section 6.11, for an example). Note that, in this human evaluation, we implemented careful study design protocols to minimize potential priming. Particularly, to ensure objectivity in the results, no information was provided about the source of the selected sentences.

There were 25 university students and staff (i.e., lecturers, professors, engineers, research scientists, and research fellows, representing diverse professional backgrounds, education levels, career stages, and age groups) participating in our survey. All participants reported using email for work and study, and have both experienced and heard about phishing attacks. Based on the responses, in summary, as depicted in Figure 2, 81% of participants selected either "Agree" (55%) or "Strongly Agree" (26%) when asked if they believe the selected sentences affect users' decision to follow the instructions in the survey phishing emails (note that in each phishing email, we use the top-1 selected sentence obtain from our method). In contrast, 10% of participants chose "Neutral", while 9% chose "Disagree" and "Strongly Disagree".

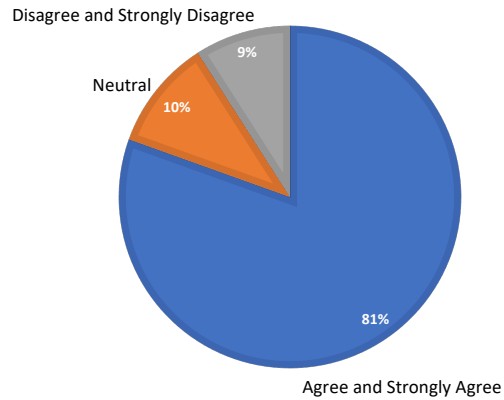

Figure 2: Human evaluation on the importance of the top-1 selected information (i.e., a sentence) from each email (by our AI2TALE method) in affecting and persuading users to follow the instructions from the email. We evaluate the selected sentences of 10 different phishing emails (randomly chosen from the testing set).

The human evaluations demonstrate the effectiveness of our method in successfully identifying the critical phishing information (reflecting the way attackers use it to deceitfully persuade users). The selected information provides a useful and concise explanation of the vulnerability prediction of the email, especially for the phishing one, to users. In essence, our approach enhances the explainability (transparency) of phishing detection, addressing and improving the overall clarity of the prediction process.

## 5  CONCLUSION

In this paper, we have successfully proposed an innovative deep learning-based method derived from an information-theoretic perspective and information bottleneck theory for solving the phishing attack localization problem where automated AI-based techniques have not yet been well studied. Our AI2TALE method works effectively in a weakly supervised setting, providing a practical solution that not only accurately predicts the vulnerability of the email data but also has the capability to automatically identify the most important and phishing-relevant information in each phishing email. The selected information provides useful and concise explanations about the vulnerability of the associated phishing email data. In addition, we also introduce appropriate measures for phishing attack localization. The rigorous and comprehensive experiments on seven real-world diverse email datasets show the superiority of our proposed AI2TALE method over the state-of-the-art baselines.

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

## 6 APPENDIX

### 6.1 REPARAMETERIZATION FOR CONTINUOUS OPTIMIZATION

Recall that the selected sentences $\tilde{X}$ are constructed by $\tilde{X} = \mathbf{z} \odot X$ where $\mathbf{z} \sim \text{MultiBernoulli}(\mathbf{p})$ with $\mathbf{p}$ is the output of the selection process $\zeta$ parameterised by a neural network $\omega(., \alpha)$.

To make this computational process (i.e., the process consists of sampling operations from a Multi-Bernoulli distribution) continuous and differentiable during training, we apply the temperature-dependent Gumbel-Softmax trick (Jang et al., 2016; Maddison et al., 2016) for relaxing each Bernoulli variable $z_i$. We sample $z_i(X; \alpha) \sim \text{Concrete}(\omega_i(X; \alpha), 1 - \omega_i(X; \alpha))$:

$$z_i(X; \alpha) = \frac{\exp\{(\log \omega_i + a_i)/\tau\}}{\exp\{(\log \omega_i + a_i)/\tau\} + \exp\{(\log(1 - \omega_i) + b_i)/\tau\}}$$

where we denote $\omega_i(X; \alpha)$ as $\omega_i$ while $\tau$ is a temperature parameter (i.e., that allows us to control how closely a continuous representation from a Gumbel-Softmax distribution approximates this from the corresponding discrete representation from a discrete distribution (e.g., the Bernoulli distribution)), random noises $a_i$ and $b_i$ independently drawn from **Gumbel** distribution $G = -\log(-\log u)$ with $u \sim \textbf{Uniform}(0, 1)$.

### 6.2 DATA PROCESSING AND EMBEDDING

We observed that most of the email data from the studied datasets (i.e., IWSPA-AP[1], Nazario Phishing Corpus[2], Miller Smiles Phishing Email[3], Phish Bowl Cornell University[4], Fraud emails[5], Cambridge[6], and Enron Emails[7]) are stored in the form of webpages on public websites. We preprocessed the datasets before using them for the training and testing processes such as removing non-ASCII characters. We also removed duplicated emails in the used datasets as well as similar emails (when they are in the same label category, i.e., phishing or benign). In the end, we utilized around 40,000 email data samples, where half of them are phishing, from the datasets to conduct our comprehensive and rigorous experiments. Note that the email data samples used in our experiments cover a wide spectrum of writing styles, tones, and structures (e.g., subject, content, html links, and signature) commonly encountered in email communications (e.g., formal business correspondences, informal personal messages, transactional emails (e.g., order confirmations or account notifications), marketing emails promoting products or services, and automated responses to specific actions or events).

We further preprocessed the datasets before injecting them into deep neural networks of our AI2TALE method and baselines. In the context of our paper, we viewed each email as a sequence of sentences and aim to learn and identify the most important sentence in each email contributing to the email's vulnerability label (phishing or benign). We embedded sentences of each email into vectors. For instance, consider the following sentence *"Your account has been suspended."*, to embed this sentence, we tokenized it to a sequence of tokens including *"your"*, *"account"*, *"has"*, *"been"*, *"suspended"*, and *"."* using the common Natural Language Toolkit (NLTK). We then used a 150-dimensional token Embedding layer followed by a Dropout layer with a dropped fixed probability $p = 0.2$, a 1D convolutional layer with the filter size 150 and kernel size 3, and a 1D max pooling layer to encode each sentence. Finally, a mini-batch of emails in which each email consisting of $L$ encoded sentences was fed to our proposed AI2TALE method and the baselines.

In our paper, the length of each email is padded or truncated with $L = 100$ sentences (i.e., we base on the quantile values of the emails' length of the used datasets to decide the length of each email). We observe that almost all important information relevant to the phishing vulnerability lies in the 100 first sentences or even lies in some very first sentences.

---

[1]https://github.com/BarathiGanesh-HB/IWSPA-AP/tree/master/data/
[2]https://monkey.org/~jose/phishing/
[3]http://www.millersmiles.co.uk/archives.php
[4]https://it.cornell.edu/phish-bowl
[5]https://www.kaggle.com/datasets/rtatman/fraudulent-email-corpus
[6]A private dataset
[7]https://www.cs.cmu.edu/~enron/

## 6.3 Measures

For the **Cognitive-True-Positive** metric, to measure if the top-1 selected sentence obtained from each method reflects the human cognitive principles (triggers) used in each phishing email, we are particularly based on the keywords and phrases often used in each cognitive principle. To obtain the keywords and phrases, for each cognitive principle, we first base on its definition (descriptions) pioneered in the well-known work (Cialdini, 1984) (widely cited and used in phishing-related studies (Akbar, 2014; Butavicius et al., 2015; Ferreira et al., 2015; Heijden & Allodi, 2019)). We then use ChatGPT (OpenAI et al., 2024) to obtain all the possible keywords and phrases related to each cognitive principle's definition (descriptions). For example, to the Scarcity principle, some of the related keywords and phrases can be relevant to Time-sensitive language (e.g., "Act now" or "Expires soon"), Limited availability (e.g., "Only a few left"), and Threats of consequences (e.g., "Will be deleted" or "Will lose access"). By relying on the definition (descriptions) of each cognitive principle with the help of ChatGPT in finding all the possible and main keywords and phrases, related to the definition (descriptions) of each cognitive principle, used in the Cognitive-True-Positive measure, we ensure this process is reliable and objective.

## 6.4 Baseline methods

The baselines of our method are recent, popular, and state-of-the-art interpretable machine learning approaches falling into the category of intrinsic interpretable models including L2X (Chen et al., 2018), INVASE (Yoon et al., 2019), ICVH (Nguyen et al., 2021), VIBI (Bang et al., 2021), and AIM (Vo et al., 2023b) that we apply to solve the email phishing attack localization problem. As mentioned in Section 2, intrinsic interpretable machine learning techniques, a.k.a. self-explanatory models, are strongly suitable for phishing attack localization because they can not only make predictions themselves but also identify the most important features of the data obtained from the model to explain the model's predictive decision.

We briefly summarize the baselines as follows:

- **L2X** (Chen et al., 2018). An efficient instance-wise feature selection method leverages mutual information for model interpretation. L2X aims to extract a subset of features that are most informative for each given example and the associated model prediction response.

- **INVASE** (Yoon et al., 2019). Another effective approach for instance-wise feature selection. INVASE consists of three neural networks including a selector network, a predictor network, and a baseline network which are used to train the selector network using the actor-critic methodology.

- **ICVH** (Nguyen et al., 2021). One of the first interpretable deep learning-based methods applied for source code vulnerability localization. ICVH is based on mutual information and a multi-Bernoulli distribution selection process for selecting vulnerability-relevant source code statements.

- **VIBI** (Bang et al., 2021). VIBI is a system-agnostic method providing a brief and comprehensive explanation by adopting an information-theoretic principle, the information bottleneck principle, as a criterion for finding such explanations.

- **AIM** (Vo et al., 2023b). A recent innovative additive instance-wise framework for model interpretation. AIM integrates both feature attribution (producing relative importance scores to each feature) and feature selection (directly identifying the subset of features most relevant to the model behavior being explained) into an effective framework for multi-class model interpretation.

It is worth noting that our proposed AI2TALE method and the baselines can work on the weakly supervised setting where during the training process, we only utilize the vulnerability label of the data, without requiring the ground truth of phishing information in the data for the task of phishing attack localization.

## 6.5 Model's configuration

For the **L2X** (Chen et al., 2018), **ICVH** (Nguyen et al., 2021), **VIBI** (Bang et al., 2021), and **AIM** (Vo et al., 2023b) methods, they were proposed to explain the output of black-box learning models. To use these methods for phishing attack localization, we keep their principles and train the models to directly approximate $p(Y|X)$ using $p(Y|\tilde{X})$ where $\tilde{X}$ consists of the selected sentences while $Y$ is ground truth label of the data $X$ instead of the output from the black-box model.

Note that the L2X, ICVH, VIBI, and AIM methods were also proposed to work with sequential text data. For these methods, we follow the structures outlined in the corresponding original papers for the architecture of the selection network that obtains $\tilde{X}$, as well as the classifier working on $\tilde{X}$ to mimic $p(Y|X)$, using the same suggested value ranges for hyper-parameters. For the INVASE method, originally designed for tabular data, we adapt it to be applicable to sequential text data by retaining its principles and using the same data embedding technique (also used in our method and other baselines) and the selection network architecture as in our AI2TALE method.

We implemented our AI2TALE method in Python using Tensorflow (Abadi et al., 2016). The trade-off parameter $\lambda$ is in $\{10^{-1}, 10^{-2}, 10^{-3}\}$ while $\sigma$ is in $\{10^{-1}, 2 \times 10^{-1}, 3 \times 10^{-1}\}$. For the networks $\omega(\cdot; \alpha)$ and $f(\cdot; \beta)$, we used deep feed-forward neural networks having three and two hidden layers with the size of each hidden layer in $\{100, 300\}$, respectively. The dense hidden layers are followed by a ReLU function as nonlinearity and Dropout (Srivastava et al., 2014) with a retained fixed probability $p = 0.8$ as regularization. The last dense layer of the network $\omega(\cdot; \alpha)$ for learning a discrete distribution is followed by a sigmoid function while the last dense layer of the network $f(\cdot; \beta)$ is followed by a softmax function for predicting. The temperature $\tau$ for the Gumbel softmax distribution is in $\{0.5, 1.0\}$. Note that we utilized the commonly used values for these hyper-parameters.

For our AI2TALE method and baselines, we employed the Adam optimizer (Kingma & Ba, 2014) with an initial learning rate of $10^{-3}$, while the mini-batch size is 128. We split the data set into three random partitions. The first partition contains 80% for training, the second partition contains 10% for validation and the last partition contains 10% for testing. We used 10 epochs for the training process. We additionally applied gradient clipping regularization to prevent over-fitting. For each method, we ran the corresponding model several times and reported the averaged **Label-Accuracy** and **Cognitive-True-Positive** measures.

We ran our experiments on a 13th Gen Intel(R) Core(TM) i9-13900KF having 24 CPU Cores at 3.00 GHz with 32GB RAM, integrated Gigabyte RTX 4090 Gaming OC 24GB.

## 6.6 Additional quantitative results

**Model Performance on F1-Score, False Positives, and False Negatives** In our study, via the Label-Accuracy metric, we aim to measure whether the selected sentences obtained from each model are truly important and help accurately predict the associated emails' vulnerability label (i.e., phishing or benign). The underlying intuition is that the most pivotal sentences significantly influence the vulnerability of emails, particularly phishing ones. Specifically, in the experiments, for each email, we focus on evaluating the most important (top-1) selected sentence (i.e., aiming to provide the most highly qualified and concise explanation of the vulnerability of email data to users) from each model to determine its efficacy in predicting the email's vulnerability label, without considering all of its sentences. *The higher the value of the Label-Accuracy measure, the better the performance in selecting the most crucial and label-relevant information from the data.*

To examine our experiments further on this aspect, we calculated the F1-score for our AI2TALE method and baselines. Specifically, we computed the F1-score in two scenarios: (i) exclusively on the testing phishing emails, and (ii) on both the testing phishing and benign emails. The results for our AI2TALE method and the baselines (i.e., AIM, VIBI, L2X, ICVH, and IN-VASE) in scenarios (i) and (ii) are shown in Table 3. In this table, we also compared the false positive rate and the false negative rate for our AI2TALE method and the baselines.

In summary, the F1-score results for our AI2TALE method and baselines closely align with those obtained using the accuracy measure (i.e., Label-Accuracy). Notably, our AI2TALE method exhibits consistent effectiveness and advancement, considerably surpassing the baselines in both sce-

narios (i) and (ii). Additionally, our AI2TALE method achieves the best performance in both false positive and false negative rates, further highlighting its superiority over the baselines.

Table 3: The performance of our AI2TALE method and the baselines for the F1-score measure on the testing set in two scenarios (i) and (ii). We also present the performance of our AI2TALE method and the baselines for the false positive rate (FPR) and false negative rate (FNR) (the best results in **bold**).

| Methods | F1-score (i) ↑ | F1-score (ii) ↑ | FPR ↓ | FNR ↓ |
|---|---|---|---|---|
| INVASE (Yoon et al., 2019) | 98.313% | 98.299% | 2.353% | 1.048% |
| ICVH (Nguyen et al., 2021) | 96.732% | 96.725% | 3.355% | 3.195% |
| L2X (Chen et al., 2018) | 98.261% | 98.249% | 2.253% | 1.248% |
| VIBI (Bang et al., 2021) | 96.626% | 96.649% | 2.504% | 4.194% |
| AIM (Vo et al., 2023b) | 98.406% | 98.399% | 1.853% | 1.348% |
| AI2TALE (Ours) | **99.324%** | **99.325%** | **0.451%** | **0.899%** |

**Data representation learning using a transformer-based large language model**  In our study, we treat each email as a sequence of sentences, where each sentence is a sequence of words (tokens), and aim to identify the most important sentence contributing to the email's vulnerability label (phishing or benign). For tokenization, we use the Natural Language Toolkit (NLTK), a widely adopted tool for natural language processing tasks.

Inspired by the baselines, to encode each sentence, we used a 150-dimensional token Embedding layer followed by a Dropout layer with a dropped fixed probability $p = 0.2$, a 1D convolutional layer with the filter size 150 and kernel size 3, and a 1D max pooling layer. The combination of the Embedding layer, 1D convolutional layer, and 1D max pooling layer is known to be a lightweight (resource-efficient) and effective approach for processing text data and learning representations. Notably, the Embedding and 1D convolutional layers are learnable during training. *We refer to this method as E1D.*

In addition to the E1D method for text data representation learning, another solution is to leverage the power and structure of a transformer-based pre-trained large language model (e.g., BERT (Devlin et al., 2019) or T5 (Raffel et al., 2020)), *particularly via its encoder component, which is specialized in data representation learning*. In this section, we use the encoder component of the BERT model for the data representation learning part of our AI2TALE method. BERT, which stands for Bidirectional Encoder Representations from Transformers, is relatively lightweight in terms of computational resources compared to many other transformer-based LLMs, like T5. BERT was designed to pre-train deep bidirectional representations from unlabeled text by jointly conditioning on both left and right context in all layers. It has achieved state-of-the-art results on various natural language processing tasks, including natural language inference, question answering, and classification. *We refer to this method as EBERT.* Note that in the case of using EBERT, during the training process, the encoder will also be fine-tuned, allowing it to adapt to the specific data and task at hand. The outputs from EBERT are the token-level representations of the input, which are then processed to form sentence-level representations. These sentence representations will be passed on to the next part of our AI2TALE model. In terms of computing resources for this case, we use a Linux-based x86-64 machine (Precision 7865 Tower) with an AMD Ryzen Threadripper PRO 5955WX and two RTX 6000 Ada Generation GPUs, each with 48 GB of VRAM.

The experimental results of our AI2TALE method for the E1D and EBERT cases are (99.33%, 98.95%, 0.451%, and 0.899%) for Label-Accuracy, Cognitive-True-Positive, FPR, and FNR in the E1D case, and (97.25%, 97.15%, 2.303%, and 3.195%) for the same metrics in the EBERT case.

In short, the E1D method is significantly more lightweight than the EBERT method. Moreover, the performance of our AI2TALE model using E1D substantially outperforms that of EBERT. We observed that this discrepancy can be attributed to several potential factors: (i) Domain shift: Differences between the domain of the pre-training data and the downstream task domain can hinder transferability; (ii) Task complexity: Some downstream tasks may be more complex or have patterns that differ from those of the pre-training objectives. In such cases, the model may not fully adapt to the unique demands of the task; (iii) Lack of task-specific knowledge: While LLMs are trained on vast datasets, they may lack specialized knowledge about the nuances of certain tasks. Although fine-tuning can help, it might not be sufficient for highly specialized or domain-specific tasks.

In our future work, we will explore innovative fine-tuning strategies and appropriate modifications to harness the capabilities of LLMs, combined with the principles of our AI2TALE method, to tackle the email phishing attack localization challenge, with the goal of identifying optimal solutions.

## 6.7 ADDITIONAL QUALITATIVE RESULTS

In Table 4, we expand the qualitative results presented in Table 2. In particular, to further aid in label explanation, along with the most important (top-1) selected sentence (which provides concise and useful information), we also present the second and third most relevant sentences for each email. It is evident that these selected sentences complement each other in elucidating the predicted label, offering additional insights to help users better understand the phishing attacks.

In Table 5, we further show qualitative experimental results of our AI2TALE method on some additional phishing email samples. Via these qualitative results, our method again demonstrates *its effectiveness in learning and identifying the most important and phishing-relevant information (i.e., sentences) from phishing emails for providing useful and concise interpretations corresponding to the predicted phishing labels*. Analysis of these phishing email samples reveals that the most important (top-1) selected sentences (as well as the second and the third selected sentences) from the model primarily reflect commonly used cognitive triggers related to the principles of scarcity (i.e., *creates a sense of scarcity and urgency*), authority (i.e., *creates a sense of trust and legitimacy*), and consistency (i.e., *creates a sense of familiarity and consistency*). Additionally, we observe that the most important sentence contributing to a specific phishing email tends to appear in the initial positions of the email to capture the users' attention.

## 6.8 MODEL SENSITIVITY TO HYPERPARAMETER CHANGES AND THE IMPACT OF INFORMATION BOTTLENECK AND DATA-DISTRIBUTION MECHANISMS

In this section, we first examine the sensitivity of our AI2TALE method to the hyperparameters used in the mutual information maximization (i.e., $\max(\mathbf{I}(\tilde{X}, Y) - \lambda \mathbf{I}(X, \tilde{X}))$) described in Eq. (8). In particular, we explore the impact of varying values for the trade-off $\lambda \in \{10^{-1}, 10^{-2}, 10^{-3}\}$ and $\sigma \in \{10^{-1}, 2 \times 10^{-1}, 3 \times 10^{-1}\}$, which are commonly used in such experiments. The results show that our AI2TALE method exhibits stability with small variances in key performance metrics, including Label-Accuracy (0.0124), Cognitive-True-Positive (0.0852), and F1-score (0.0123). Achieving high performance, around 99.33% in both Label-Accuracy and F1-score, and 98.95% in Cognitive-True-Positive, along with small variances, indicates the robustness of our AI2TALE method.

In our AI2TALE method, although the joint training process mentioned in the main paper via the Eq. (1) guided by information theory (through mutual information) brings benefits for selecting the important and phishing-relevant sentences from phishing emails, we observe two potential limitations of this training principle, including obtaining a superset of phishing-relevant sentences, and encoding the vulnerability label via its selections instead of via truly meaningful information. To tackle these potential issues, we propose two innovative training solutions: (i) information bottleneck theory training term that ensures only important and label-relevant information (i.e., sentences) will be kept and selected, and (ii) data-distribution mechanism that ensures the learnable selection process respecting the data distribution to select the meaningful and label-relevant information. This mechanism also aids in improving the generalization capability of the selection network.

We examine the impact of the information bottleneck theory training term and data-distribution mechanism on the model performance. Without these terms, our model's performance matches the second-highest baseline in Cognitive-True-Positive, which is ICVH. We observe improvements of approximately 0.5% in Label-Accuracy and 0.8% in False Positive Rate (FPR) compared to the second-highest baseline (AIM). However, there is an increase of about 0.4% in False Negative Rate (FNR) compared to the second-highest baseline (INVASE). When these terms are applied, our AI2TALE method consistently outperforms the baselines, achieving the best performance across all metrics, including Label-Accuracy (99.33%), Cognitive-True-Positive (98.95%), F1-score (99.33%), False Positive Rate (0.451%), and False Negative Rate (0.899%) by a wide margin. In this case, our method shows an improvement of approximately 1.5% to 3.5% in the average of two key metrics (Label-Accuracy and Cognitive-True-Positive) compared to the baselines.

## 6.9 THREATS TO VALIDITY

**Construct validity**   Key construct validity threats are if the assessments of our AI2TALE method and baselines demonstrate the ability for phishing attack localization. In our paper, we study an important problem of phishing attack localization where we not only aim to automatically learn and identify the most important and phishing-relevant information (e.g., sentences) in each phishing email but also can detect the vulnerability label $Y$ (i.e., phishing or benign) of the corresponding email based on the crucial selected information. The selected phishing-relevant information (e.g., sentences) helps provide useful and concise explanations about the vulnerability of the phishing email data. To evaluate the performance of our method and baselines, we utilize two main measures including Label-Accuracy and Cognitive-True-Positive, supported by additional measures such as F1-score, False Positive Rate, and False Negative Rate.

**Internal validity**   Key internal validity threats are relevant to the choice of hyperparameter settings, such as the optimizer, the learning rate, and the number of layers in deep neural networks. It is worth noting that finding a set of optimal hyperparameter settings of deep neural networks is expensive due to a large number of trainable parameters. To train our method, we chose to use the common and default values of hyperparameters, e.g., using Adam optimizer and the learning rate of $10^{-3}$. We also report the hyperparameter settings in our released reproducible source code samples to support future replication studies.

**External validity**   Key external validity threats include whether our proposed AI2TALE method can generalize well to different types of phishing email datasets. We mitigate this problem by conducting our experiments on seven real-world diverse email datasets including IWSPA-AP, Nazario Phishing Corpus, Miller Smiles Phishing Email, Phish Bowl Cornell University, Fraud emails, Cambridge, and Enron Emails.

## 6.10 PRIVACY CONCERNS AND THE RISK OF MISCLASSIFICATION

Privacy is an important concern in AI and machine learning systems, especially when dealing with sensitive personal data. Training AI models on large datasets carries the risk of inadvertently exposing private information. This can occur through direct access to the training data or through model inference, where the model may unintentionally memorize and reveal private details.

Regarding privacy concerns when deploying our AI2TALE method in real-world applications, the operations of our AI2TALE method and its approach to addressing the phishing attack localization problem are designed to minimize these risks. In particular, the explainable component of our method allows users to understand the reasoning behind flagged phishing attempts by highlighting the most important and label-relevant information directly from the data tested along with the predicted label. Importantly, this process does not expose any sensitive data from the training set, minimizing the risk of privacy breaches throughout.

In terms of misclassification issues, when the model produces an incorrect prediction about the vulnerability label (phishing or benign) of the email, experiments on seven diverse real-world email datasets show that our AI2TALE method achieves around 99.33% for both Label-Accuracy and F1-score, with a low False Positive Rate of 0.451% and a low False Negative Rate of 0.899%. This indicates that the rate of misclassifications is minimal. In the case of false positives, the highlighted information from the data still provides users with useful insights, enabling them to take proactive measures to enhance security. For false negatives, the highlighted information also allows users to double-check the flagged content (if they wish), minimizing the likelihood that potential threats are overlooked. In conclusion, the predictions and explanations work together to provide users with interconnected, supportive information. Furthermore, by highlighting the most important and label-relevant information, the model facilitates users in receiving clear, actionable insights.

## 6.11 EXAMPLE USED IN HUMAN EVALUATION

In the human evaluation, we evaluate whether the most important (top-1) selected sentence in each phishing email by our proposed AI2TALE method is perceived as convincing information for email users. To do that, we asked participants to evaluate the selected sentences of 10 different phishing emails (randomly chosen from the testing set) in terms of whether the sentence selected in each

email is important to influence and persuade users to follow the instructions in the emails. Below, we present an example question.

"Given an email as follows, do you think the selected sentence (in bold) affects and persuades users' decision to follow the instructions from the email?

Bulk attention! Your discover account will close soon! **Dear member, we have faced some problems with your account, so please update the account**. If you do not update will be closed. To update your account, just confirm your information. (it only takes a minute). It's easy. 1. Click the link below to open a secure browser window. 2. Confirm that you're the owner of the account, and then follow the instructions.

The participants will then choose one option from the following: Strongly Agree, Agree, Neutral, Disagree, or Strongly Disagree."

### 6.12 FUTURE WORK

In this study, our paper focuses primarily on addressing the phishing attack localization problem in email phishing, the most common form of phishing. It is worth noting that phishing attacks can also occur through webpages. Several methods have been proposed to detect and infer phishing intention based on webpage appearances (e.g., (Abdelnabi et al., 2020; Liu et al., 2022)). We believe the principles underlying our AI2TALE method can also be extended to *detect and localize* webpage phishing attacks. The operational nature of our AI2TALE method within (i) a unified framework, (ii) working directly with webpage data without requiring additional steps to gain extra information (e.g., references to the ground truth of phishing information) beyond the data and its vulnerability label (i.e., phishing or benign), as well as (iii) the way our AI2TALE model can be trained without requiring the ground truth of phishing information in the data can be considered as some of the advantages of our method compared to the relevant methods (e.g., (Abdelnabi et al., 2020; Liu et al., 2022)). Investigating the application of our AI2TALE method to explain webpage phishing attacks could be a focus of our future studies.

### 6.13 RELATED BACKGROUND

Following, we briefly present the main related background used in our proposed AI2TALE method.

**Mutual information**    Mutual information (MI) is used to measure the dependence between two random variables (Shannon, 1998; Cover & Thomas, 2006). It captures how much the knowledge of one random variable reduces the uncertainty of the other. In particular, MI quantifies the amount of information obtained about one random variable by observing the other random variable. For instance, suppose variable A signifies the outcome of rolling a standard 6-sided die, and variable B represents whether the roll yields an even (0 for even, 1 for odd) result. It is evident that information from B offers insights into the value of A, and vice versa. In essence, these random variables exhibit mutual information.

Assume that we have two random variables $X$ and $Y$ drawn from the joint distribution $p(x, y)$ with two corresponding marginal distributions $p(x)$ and $p(y)$. The mutual information between $X$ and $Y$ denoted by $I(X, Y)$ is the relative entropy between the joint distribution $p(x, y)$ and the product distribution $p(x)p(y)$, and is defined as follows:

$$I(X, Y) = \int p(x, y) \log \frac{p(x, y)}{p(x)p(y)} dx dy = D_{KL}(p(x, y) || p(x)p(y))$$

where $D_{KL}(p(x, y) || p(x)p(y))$ is the Kullback-Leibler divergence measuring the relative entropy (i..e, the difference in information) represented by two distributions, i.e., the product of marginal distributions $p(x)p(y)$ of $X$ and $Y$ from their joint distribution $p(x, y)$.

**Information bottleneck theory**    Here, we consider the supervised learning context where we want to predict corresponding outputs (e.g., labels) $\{\mathbf{y}_i\}_{i=1}^{n}$ of given inputs $\{\mathbf{x}_i\}_{i=1}^{n}$. A deep learning network (DNN) will learn latent representations (i.e., latent features in the latent space that contain

useful information to describe the data) $\{\tilde{\mathbf{x}}_i\}_{i=1}^n$ of the corresponding input data samples $\{\mathbf{x}_i\}_{i=1}^n$ in terms of enabling good predictions and generalizations.

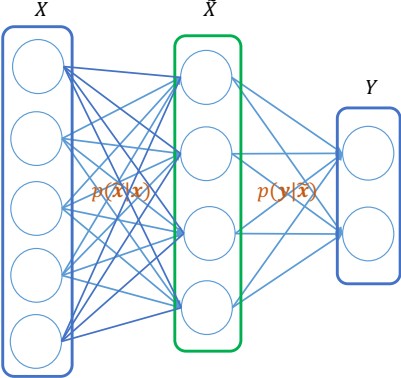

Figure 3: An architecture of a simple deep neural network in a supervised learning context for the classification problem.

Assume that the whole hidden layer in Figure 3 is denoted by a random variable $\tilde{X}$ while the input and output layers are denoted by random variables $X$ and $Y$ respectively. We can describe this hidden layer by two conditional distributions: the encoder $p(\tilde{\mathbf{x}}|\mathbf{x})$ and the decoder $p(\mathbf{y}|\tilde{\mathbf{x}})$. This transformation process preserves the information of the input layer $X$ without considering which individual neurons within the hidden layer $\tilde{X}$ encode which features (i.e., neurons) of $X$. An optimal encoder process of the mutual information between $X$ and the desired output $Y$ denoted by $I(X, Y)$ can create the most compact encoding (i.e., minimally sufficient statistic) $\tilde{X}$ of the input data $X$ while $\tilde{X}$ still has enough information (i.e., $\tilde{X}$ can capture the important features of $X$ as well as remove the unnecessary parts of $X$ that do not make contributions to the prediction of $Y$) to predict $Y$ as accurately as possible.

An information bottleneck (Tishby et al., 2000; Naftali Tishby, 2015) is proposed to be a computational framework that aims to find the most compact encoding $\tilde{X}$ of the input data $X$. In particular, it is the optimal trade-off between the compression $\tilde{X}$ and the prediction of the desired output $Y$ as described in the following optimization problem:

$$\min_{p(\tilde{\mathbf{x}}|\mathbf{x}), p(\mathbf{y}|\tilde{\mathbf{x}}), p(\tilde{\mathbf{x}})} \left\{ I(X, \tilde{X}) - \beta I(\tilde{X}, Y) \right\}$$

where $\beta$ specifies the amount of relevant information captured by the encoding process (i.e., the representations $\tilde{X}$ and $I(\tilde{X}, Y)$).

Table 4: The ground truth and the predicted label based on the most important (top-1) selected sentence (in yellow) of each email from our AI2TALE method. The top-1 selected sentence provides concise and useful information. However, to further aid in label explanation, here, we also show the second (in melon) and third (in pink) most relevant sentences.

| Truth | Model | Emails |
| --- | --- | --- |
| phishing | phishing | **The email:** "Please confirm your online banking records! Dear customer of natwest bank, we are running a scheduled maintenance on our servers. We want to make sure your money and your personal details are safe and secure. Due to new security policies all natwest bank customers must complete the natwest customer form. To complete the form, please use the link below natwest customer form this should take you directly to the natwest customer form. Sincerely, natwest customer service good." 
 **Note:** The phishing email poses as a legitimate message from natwest bank, requesting confirmation of banking records. Its goal is to trick individuals into providing personal information under the guise of security, emphasizing the need to verify the authenticity of such communications. Our method successfully figures out the sentence "Please confirm your online banking records!" representing the key message of the phishing email. It serves to create urgency and prompt the recipient to take action, suggesting that there is a need to verify their record information. 
 In addition, the second and third selected sentences, "Due to new security policies, all natwest bank customers must complete the natwest customer form." (in melon) and "We want to make sure your money and your personal details are safe and secure." (in pink), respectively create a sense of urgency and authority, as well as evoke feelings of warmth and security, subtly reassuring the recipient and lowering their guard. |
| phishing | phishing | **The email:** "Temporarily suspended. Dear customer, customer advice, please address the following issues. The details that you have entered have not been recognized. For your security, your online service has been temporarily locked. No further attempts will be accepted. If you provide us with the following details, you should be able to access the service in just a few minutes. Click here to get started legal info privacy security 2005 2010." 
 **Note:** This phishing email tries to convey (i) the action taken is intended to protect the recipient's well-being and (ii) a sense of urgency, encouraging the recipient to address the issue promptly to regain access to their online service. Our method successfully figures out the sentence "For your security, your online service has been temporarily locked." representing the key message of the phishing email. 
 In addition, the second and third selected sentences, "If you provide us with the following details, you should be able to access the service in just a few minutes." (in melon) creates a sense of urgency and promises a quick resolution, prompting the recipient to act immediately without thinking critically. Meanwhile, "Dear customer, customer advice, please address the following issues." (in pink) aims to create an illusion of professionalism and authority, making the recipient feel reassured and more likely to trust the email. |
| phishing | phishing | **The email:** "Bulk attention! Your discover account will close soon! Dear member, we have faced some problems with your account, so please update the account. If you do not update will be closed. To update your account, just confirm your information. (it only takes a minute). It's easy. 1. Click the link below to open a secure browser window. 2. Confirm that you're the owner of the account, and then follow the instructions." 
 **Note:** The message from the yellow selected sentence obtained from our method exhibits cognitive triggers commonly associated with phishing attempts used in the phishing email. In particular, it implies a sense of urgency (concern) via problems with your account while "Dear member" aims to establish a connection with the recipient and imply that the message comes from a trusted source. The phrase "please update the account" creates a sense of familiarity and consistency. 
 In addition, the instruction to "just confirm your information" and the statement "It's easy" (via the second (in melon) and third (in pink) selected sentences) aim to minimize perceived effort, making the recipient more likely to comply without hesitation. |
| benign | phishing | **The email:** " Lower your mortgage payment! Bad credit, no problem! Lower your mortgage payment! Bad credit, no problem! Whether your credit is excellent or less than perfect, loanweb has a lender that can help you! Lowest rates on the web! Bad credit? Refinance to get cash! Lower monthly payments! Bad credit? Refinance to consolidate bills! Click here for lower mortgage payment! Win a free bread maker! Win free bread makers, toaster ovens, cookware and more get a free subscription to cooking pleasures magazine get a free multi purpose grater test and keep free cooking products get free recipes from world famous chefsplus get a free 90 day membership in the cooking club of america. Home equity loans without perfect credit! Click here for a home equity loan! Free application! Home equity loans up to 125 potential tax deductible interest! Customized, competitive equity lines and loans. Apply now!" 
 **Note:** In the datasets we utilized, the ground truth label for this email is benign. However, our method predicts it as phishing due to the detection of potential phishing indicators, such as the use of enticing offers of "lowering mortgage payments" and the presence of "suspicious links" potentially prompting users to provide personal information and bank account details via the phrase "Click here for lower mortgage payment!". Although the email's ground truth classification was incorrect, we deem the predicted label and the highlighted sentence from our model valuable in alerting users to potential phishing attempts. 
 In addition, the second and third selected sentences, "Refinance to consolidate bills!" (in melon) and "Lower your mortgage payment!" (in pink), are designed to appeal to individuals struggling with debt or looking for ways to manage their finances. These phrases create the impression of offering a legitimate and helpful financial product, while also aiming to immediately grab the recipient's attention. |

Table 5: The ground truth and the predicted label based on the most important (top-1) selected sentence (in yellow) of each email from our AI2TALE method. The top-1 selected sentence provides concise and useful information. However, to further aid in label explanation, here, we also show the second (in melon) and third (in pink) most relevant sentences.

| Truth | Model | Emails |
|---|---|---|
| phishing | phishing | **The email:** "Dear paypal customer. Dear paypal customer, this is an official notification from paypal that the service listed below will be deactivated and deleted if not renewed immediately. Previous notifications have been sent to billing contact assigned to this account. As the primary contact, you must renew the service listed below, or it will be deactivated and deleted. Click to renew your paypal account now service paypal security department. Expiration may 14, 2010, at paypal we are dedicated to providing you with exceptional service and to ensuring your trust. If you have any questions regarding our services, please check the website or call our customer service. Thank you, sincerely, paypal security department paypal's services terms and conditions apply. The information on this page is presented subject to our legal page and any other terms and conditions that paypal may impose from time to time. It is subject to change without notification. Microsoft and the microsoft internet explorer are registered trademarks or trade works of microsoft corporation in the united states and for other countries." 
 **Note that:** The yellow selected sentence employs urgent (fear and concern) language by stating that the service will be "deactivated and deleted if not renewed immediately". This urgency can create panic and pressure the recipient to act quickly without considering the legitimacy of the message. Furthermore, it also attempts to establish credibility and authority via "official notification from paypal" often used by attackers. The message from the selected sentence starts with the generic address "Dear paypal customer" twice. While legitimate communications would typically use the recipient's actual name, phishing emails often lack personalization and use general greetings. In general, the selected sentence consists of almost all key messages from the email. That shows the effectiveness of our proposed AI2TALE method in identifying the most important phishing-relevant information (sentence) from the email. 
 In addition, the second and third selected sentences, "Click to renew your paypal account now, service paypal security department" (in melon) and "Thank you, sincerely, paypal security department paypal's services terms and conditions apply" (in pink), both create urgency, mimic authority, and use vague or formal language to gain trust. The melon sentence pressures the recipient to click a link, while the pink sentence attempts to establish credibility with official-sounding language. |
| phishing | phishing | **The email:** "Your payment didn't succeed, so your ads have been suspended. This message was sent from a notification only email address that does not accept incoming email. Please do not reply to this message. If you have any questions, please visit the google ads help center. Hello advertiser, our attempt to charge your credit card for your outstanding google ads account balance was declined. Your account is still open. However, your ads have been suspended. Once we are able to charge your card and receive payment for your account balance, we will re activate your ads. Please update your billing information, even if you plan to use the same credit card. This will trigger our billing system to try charging your card again. You do not need to contact us to reactivate your account. To update your primary payment information, please follow these steps 1. Log in to your account at http ad words google com select. 2. Enter your primary payment information. 3. Click "update" when you have finished. Thank you for advertising with google ads. We look forward to providing you with the most effective advertising available. (c) google ads team 2008." 
 **Note:** The yellow selected sentence "Your payment didn't succeed, so your ads have been suspended." exemplifies key tactics in the phishing email by creating a sense of urgency and alarm, prompting recipients to act quickly. It impersonates legitimate Google Ads communications to gain trust and includes a call to action to update billing information, aiming to harvest personal and financial details. By conveying authority and legitimacy, the message seeks to distract recipients from potential red flags, increasing the likelihood of falling victim. 
 In addition, the second and third selected sentences, "Hello advertiser, our attempt to charge your credit card for your outstanding google ads account balance was declined" (in melon) and "Please update your billing information, even if you plan to use the same credit card" (in pink), use authoritative language to appear legitimate and pressure the recipient into acting without careful consideration. The melon sentence exploits the fear of an account issue, while the pink sentence suggests a specific action ("update billing information"), a common phishing tactic to collect sensitive data. |

