# OpenReview forum: "AI2TALE: An Innovative Information Theory-based Approach for Learning to Localize Phishing Attacks"
_ICLR.cc/2025/Conference — ICLR 2025 Poster_

### Official Review · Reviewer_YZX3 · 2024-10-23

**Soundness:** 3
**Presentation:** 3
**Contribution:** 3
**Rating:** 8
**Confidence:** 4

**Summary:**

This paper presents a novel approach to phishing attack localization using deep learning and information theory principles. The method aims to classify emails as phishing or benign and explain these classifications by identifying the most relevant information within the emails. In short, phishing attack localization aims to customize phishing attacks to align with a target region or group's specific cultural, linguistic, and contextual characteristics. The paper examines how localized phishing tactics, such as using native language, regional events, or imitating local institutions, enhance the effectiveness of phishing campaigns by increasing their credibility. The study compares localized versus generic phishing attacks and highlights key factors that improve success rates. It also discusses the growing role of AI in automating localization efforts, presenting a challenge for organizations to develop stronger, region-specific defense mechanisms to counter these advanced attacks. The authors have conducted extensive experiments on multiple datasets to demonstrate the efficacy of their approach. Overall, it’s an interesting paper with a sound contribution.

**Strengths:**

Strengths:
- Original Contribution: The proposed method (enhances explainability) is a significant improvement in phishing defense and a good use of AI.
- The use of information theory and the mutual information training principle to select relevant features is well-founded. The introduction of a selection network that utilizes latent variables to identify important sentences is also good.
- The authors provide thorough evaluations across seven diverse datasets. The comparative performance metrics (Label-Accuracy and Cognitive-True-Positive) are appropriate and lend credibility to their claims of improved performance over baseline models.
- The paper is well-organized, with clear sectioning and logical flow. The introduction succinctly frames the problem and the proposed solution, making it accessible even to readers less familiar with the technical intricacies.
- The use of human evaluations to assess the interpretability of the selected sentences enhances the practical relevance of the findings.

**Weaknesses:**

Weaknesses:
- It could be good to add a more elaborate discussion of the ethical implications of deploying this AI-based system in real-world applications, particularly concerning privacy concerns and the risk of misclassification.
- The authors acknowledge potential limitations regarding selecting non-relevant sentences. A more thorough discussion of these limitations and how they might affect real-world applications would be good. For example, the risk of misclassification and potential consequences of user confusion arising from irrelevant explanations.
- The paper mentions the introduction of hyperparameters in the mutual information maximization and data-distribution mechanisms. It would be good to elaborate on the model’s sensitivity to these hyperparameters and their impact on the performance. It would also be beneficial for the authors to discuss how the introduced hyperparameters are likely to generalize to different contexts and model types. Understanding whether these hyperparameters remain effective across various datasets or when applied to different phishing detection frameworks.
- The results section could benefit from more visualizations (figures or tables) to make the results more digestible and better illustrate the performance improvements.

**Questions:**

- The paper compares the proposed method with state-of-the-art AI-based approaches. Did you consider including a more elaborate discussion of how traditional (non-AI) phishing detection methods, such as URL filtering and white and blacklisting IPs, sandboxing, etc.,  can be complemented by or replace improvements offered by AI2TALE?
- Can you elaborate on what defense mechanisms organizations should prioritize to counter the increasingly sophisticated, region-specific threats posed by localizaion?

**Details Of Ethics Concerns:**

I don't think any additional ethical review is required, but it could be good if the authors added a more elaborate discussion of the ethical implications of deploying this AI-based system in real-world applications, particularly concerning privacy concerns and the risk of misclassification.

---

> ### Author Response · Authors · 2024-11-17
> **Authors’ responses to Reviewer YZX3 (Part 1)**
>
> We appreciate the effort you have dedicated to reviewing our paper and offering insightful suggestions for further enhancements and clarifications. Your recognition of our work is greatly appreciated. Below, we answer all of the major comments and questions, and the corresponding clarifications will be integrated into the updated version of our paper.
>
> **Q.** *“The paper compares the proposed method with state-of-the-art AI-based approaches. Did you consider including a more elaborate discussion of how traditional (non-AI) phishing detection methods, such as URL filtering and white and blacklisting IPs, sandboxing, etc., can be complemented by or replace improvements offered by AI2TALE?”*
>
> **Response.** We value the suggestion to discuss how traditional (non-AI) phishing detection methods, such as URL filtering, IP blacklisting, sandboxing, etc., could be complemented or replaced by our proposed AI2TALE method.
>
> While our current work focuses on comparing deep learning-based approaches for phishing attack localization, with the aim of tackling and improving the explainability (transparency) of phishing detection (especially for email phishing), we acknowledge that traditional methods still play a crucial role in defeating phishing attacks via phishing detection. Techniques such as URL filtering, IP blacklisting, and sandboxing typically rely on predefined rules or heuristic patterns to identify phishing attempts. These methods can be highly effective at detecting known phishing sites and blocking malicious traffic based on established patterns or signatures. However, their effectiveness is often limited to recognizing known threats. As phishing techniques (data) evolve and become more sophisticated, traditional methods may struggle to keep pace.
>
> In contrast, our AI2TALE method, a deep learning-based approach, can automatically learn meaningful features from data, providing high-quality predictions and explanations (via identifying the most important, phishing-relevant information (i.e., sentences) that trigger the classification), allowing for more dynamic and adaptive solutions. As an advanced deep learning method, AI2TALE is not limited to known threats and can be effectively adapted to address new and evolving phishing techniques.
>
> Our AI2TALE method offers greater explainability, flexibility, and adaptability compared to traditional methods. We believe that, in addition to replacing them, the principles of AI2TALE can also complement traditional methods in several ways:
>
> *Building Trust in Automated Decisions.* Traditional methods like blacklisting or sandboxing can sometimes flag items inaccurately, which can lead to false positives (blocking something that is not actually malicious). AI2TALE’s explainability would help users understand why something was flagged and whether the decision is likely correct, reducing the fear of over-blocking and making administrators more comfortable trusting automated decisions. In the case of false negatives, AI2TALE’s explainability enables users to double-check flagged content, reducing the likelihood that potential threats are overlooked.
>
> *Improving Adaptability and Effectiveness.* Traditional methods often rely on predetermined rules or lists, which can be out of date or ineffective against new, evolving threats. With AI2TALE, adaptability and explainability could offer insight into new patterns or tactics, improving overall effectiveness.

---

> ### Author Response · Authors · 2024-11-17
> **Authors’ responses to Reviewer YZX3 (Part 2)**
>
> **Q.** *“Can you elaborate on what defense mechanisms organizations should prioritize to counter the increasingly sophisticated, region-specific threats posed by localization?”*
>
> **Response.** Thank you for your insightful question. As phishing attacks become more sophisticated and tailored to specific regions, we believe that organizations must adopt a proactive, multi-layered defense strategy. For example:
>
> *Technical Defenses*
>
> Leverage advanced AI-powered models to detect and explain phishing attacks. Phishing attack localization methods (e.g., our AI2TALE method) provide effective solutions. These methods not only predict the vulnerability label (i.e., phishing or benign) but also automatically identify the most important phishing-relevant information (e.g., sentences), offering concise and meaningful interpretations that provide valuable insights into the nature of the attacks.
>
> Regularly update AI-based models with new threat data to improve their ability to detect emerging phishing threats. This helps ensure that models remain effective over time.
>
> Conduct regular security assessments and penetration testing to identify vulnerabilities and ensure that both technical defenses and detection systems are functioning effectively.
>
> *Non-Technical Defenses*
>
> Employee Education and Awareness: While advanced AI-based solutions are a critical defense, employee training remains one of the most effective ways to mitigate phishing threats. Ongoing education, including phishing simulations and security awareness programs, ensures that employees can recognize phishing attempts and respond appropriately.
>
> Phishing Simulations: Regularly conduct simulated phishing attacks to help employees learn to identify suspicious emails and other phishing techniques.
>
> Security Awareness Programs: Provide continuous training on the latest phishing tactics, social engineering strategies, and safe online behaviors. This helps build a culture of vigilance across the organization.

---

> ### Author Response · Authors · 2024-11-17
> **Authors’ responses to Reviewer YZX3 (Part 3)**
>
> **Q.** *“It could be good to add a more elaborate discussion of the ethical implications of deploying this AI-based system in real-world applications, particularly concerning privacy concerns and the risk of misclassification. The authors acknowledge potential limitations regarding selecting non-relevant sentences. A more thorough discussion of these limitations and how they might affect real-world applications would be good. For example, the risk of misclassification and potential consequences of user confusion arising from irrelevant explanations.”*
>
> **Response.** The operations of our AI2TALE method and its approach to phishing attack localization are designed to minimize the risks of privacy concerns and misclassification.
>
> Regarding privacy, the explainable component of our method allows users to understand the reasoning behind flagged phishing attempts by highlighting the most important and label-relevant information directly from the data tested along with the predicted label. Importantly, this process does not expose any sensitive data from the training set, minimizing the risk of privacy breaches throughout.
>
> Regarding misclassification, experiments on seven diverse real-world email datasets show that our method achieves around 99.33% for both Label-Accuracy and F1-score, with a low False Positive rate of 0.451% and a low False Negative rate of 0.899%. This indicates that the rate of misclassifications is minimal. In the case of false positives, the highlighted information from the data still provides users with useful insights, enabling them to take proactive measures to enhance security. For false negatives, the highlighted information also allows users to double-check the flagged content (if they wish), minimizing the likelihood that potential threats are overlooked. In conclusion, the predictions and explanations work together to provide users with interconnected and supportive information. Furthermore, by highlighting the most important and label-relevant information, the model facilitates users in receiving clear and actionable insights.
>
> We have incorporated these points in Section 6.11 of the revised version of our paper.
>
> **Q.** *“The paper mentions the introduction of hyperparameters in the mutual information maximization and data-distribution mechanisms. It would be good to elaborate on the model’s sensitivity to these hyperparameters and their impact on the performance.”*
>
> **Response.** Thank you for your valuable comment.
>
> In our AI2TALE method, although the joint training process guided by information theory (through mutual information) brings benefits for selecting the important and phishing-relevant sentences from phishing emails, we observe two potential limitations of this training principle, including obtaining a superset of phishing-relevant sentences, and encoding the vulnerability label via its selections instead of via truly meaningful information. To tackle these potential issues, we propose two innovative training solutions:
>
> - *Information bottleneck theory training term*. It ensures that only important and label-relevant information (i.e., sentences) will be kept and selected.
>
> - *Data-distribution mechanism*. It ensures the learnable selection process respecting the data distribution to select the meaningful and label-relevant information. It also aids in improving the generalization capability of the selection network.
>
> Based on your suggestion, we have tested the impact of our proposed information bottleneck theory training term and data-distribution mechanism on the model performance. In short, without these terms, our model's performance is comparable to the baselines, with minor improvements across all metrics. However, when these terms are used, our AI2TALE method always achieves the best performance in terms of *Label-Accuracy, Cognitive-True-Positive, F1-score, False Positive rate, and False Negative rate*, outperforming the baselines by a wide margin.
>
> Regarding the sensitivity of our AI2TALE method to the hyperparameters in the mutual information maximization (i.e., $\max(\mathbf{I}(\tilde{X},Y)-\lambda\mathbf{I}(X,\tilde{X}))$) shown in formula (8), we explore the impact of varying values for the trade-off $\lambda \in$ {$\{10^{-1},10^{-2},10^{-3}\}$} and $\sigma \in$ {$\{10^{-1},2\times10^{-1},3\times10^{-1}\}$}, which are commonly used in such experiments. Our model demonstrates stability with small variances in key performance metrics, including Label-Accuracy (0.0124), Cognitive-True-Positive (0.0852), and F1-score (0.0123). Achieving high performance, around 99.33% in both Label-Accuracy and F1-score, and 98.95% in Cognitive-True-Positive, along with small variances, indicates the robustness of our proposed AI2TALE method.
>
> We have presented these additional assessments in Section 6.9 of the revised version of our paper.

---

> ### Author Response · Authors · 2024-11-17
> **Authors’ responses to Reviewer YZX3 (Part 4)**
>
> **Q.** *“The results section could benefit from more visualizations (figures or tables) to make the results more digestible and better illustrate the performance improvements.”*
>
> **Response.** Thank you for this suggestion. To enhance clarity and provide a more thorough comparison, along with the shown Label-Accuracy, Cognitive-True-Positive, and F1-score results, we have added a table showing the False Positive rate and False Negative rate of our AI2TALE method and the baselines in Section 6.7 of the revised version of our paper. This helps to further demonstrate the effectiveness and advancement of our method.
>
> The False Positive rate (FPR) and False Negative rate (FNR) for our AI2TALE method and the baselines are summarized as follows:
>
> INVASE (FPR: 2.353% and FNR: 1.048% )
>
> ICVH  (FPR: 3.355% and FNR: 3.195% )
>
> L2X (FPR: 2.253% and FNR: 1.248% )
>
> VIBI (FPR: 2.504% and FNR: 4.194% )
>
> AIM (FPR: 1.853% and FNR: 1.348% )
>
> AI2TALE (FPR: 0.451% and FNR: 0.899% )
>
>
> Moreover, as suggested by Reviewer PCAG (who also gave our paper a rating of 8), in addition to the top-1 selected sentence, we will also include the second and third most relevant sentences from the model. We have responded to Reviewer PCAG with an example (also in Part 4), showing the selected sentences can support each other in explaining the predicted label to users. We believe this will be beneficial for users when the method is deployed in real-world applications. This additional analysis has been updated in Section 6.8 of the revised version of our paper.

---

### Official Review · Reviewer_8txw · 2024-11-01

**Soundness:** 2
**Presentation:** 1
**Contribution:** 1
**Rating:** 3
**Confidence:** 4

**Summary:**

This paper builds a system that identifies both, whether an email text is a phishing attack and which top-1 sentence in the email has the highest contribution to the classification label. The method is based on information theory, which is used by the model to distribute importance weights to features. The paper claims that this is one of the first works in phishing localization and that XAI techniques have not been applied to phishing attacks in the literature. By comparing against 7 open source datasets and 5 popular baseline methods, the method shows an improvement of 1.5-3.5% over baselines.

**Strengths:**

It is good to focus on explainable approaches so we can understand what ML models do. The method uses information theory approaches, which is indeed an innovative angle.

**Weaknesses:**

Firstly, the claim that this is one of the only works for interpretable phishing localization might be inflated. There is plenty of work in the domain of phishing attack detection that builds ML models and applies XAI techniques to understand why something was labelled as a phishing attempt, see e.g., [1-2] as just two random examples. These techniques even go beyond text and also tag why images or URLs were considered phishing. Thus, the novelty of this approach is severely lacking. Please provide a more detailed comparison of the proposed approach to existing XAI techniques in phishing detection, highlighting specific differences in methodology or capabilities.

Second, one of the most popular explanation techniques for text, especially for deep nets is the attention mechanism. The proposed approach seems to somewhat achieve a similar goal by learning one model that both classifies text as phishing and also discovers features (sentences) that led to the classification decision. Yet, the paper makes no mention of this popular technique, nor do they compare with it in terms of fidelity, speed, correctness, etc. Thus, there is insufficient comparison with existing methods. The authors are advised to include a comparison with attention-based methods, evaluating aspects like fidelity, speed, and correctness.

Third, the writing can substantially be improved. There are several typos that hinder comprehension. A non-exhaustive list: page 2, second last para -> which information causes. page 4, first sentence -> given an email. page 5, line 255 -> we encourage.


[1] Chai, Yidong, et al. "An explainable multi-modal hierarchical attention model for developing phishing threat intelligence." IEEE Transactions on Dependable and Secure Computing 19.2 (2021): 790-803.
[2] Lin, Yun, et al. "Phishpedia: A hybrid deep learning based approach to visually identify phishing webpages." 30th USENIX Security Symposium (USENIX Security 21). 2021.

**Questions:**

Please see below some questions and comments to improve the manuscript:

- Overall, the terminology used in this paper, e.g., phishing attack localization and phishing vulnerability are atypical for the cybersecurity community. Page 1, para 2 states that AI has been used for malware vulnerability detection. Malware detection and vulnerable code detection are two entirely different sub-domains. Moreover, page 3 states that post-hoc explanations do not offer a comprehensive understanding of the model's internal architecture or workings. Overexaggerated claim. Model-based XAI techniques like integrated gradients do exactly that! Deep models are also called "self-explanatory" methods in the manuscript. Again, popular literature dictates that deep learning is inherently a black box and thus cannot be self-explanatory/interpretable. The authors are advised to revise the manuscript and avoid redefining terms that already exist in the domain.

- This work is not the first or the only work that labels an object as phishing and identifies which features led to the classification decisions. Could the authors present a comparative analysis against existing methods, e.g., [1-2], and justify why they were not mentioned in the manuscript?

- Please add references for the datasets mentioned on page 7.

- Please include the definition of the evaluation metrics (especially cognitive true positive) in the main body text (Section 4.2). Without the definition, it is impossible to understand the results.

- The significance of the results in table 1 are hard to interpret because the authors do not provide any information regarding the dataset, e.g., number of phishing emails, sentences per email etc. Moreover, the improvement in the results is marginal (1-3%). Does the proposed approach provide benefits beyond the results in Table 1?

- Please also report false positives and false negatives for each method in table 1. FPs are especially a big problem in cybersecurity tasks.

- Please provide more details regarding the human study. 25 participants don't seem statistically significant for 10 emails, each with 5 options, as estimated by Cochran's formula. Moreover, it is unclear what questions are asked or how they are phrased exactly. For instance, if the participants are always shown the top-1 sentence and not given an option of top-3 sentences, this can very likely be interpreted as a rhetorical question, which leads people to respond in the affirmative most of the time. The authors are advised to increase the sample size to a statistically significant level, provide the exact phrasing of questions asked, and include control questions or alternative options (like top-3 sentences) to reduce bias.

**Details Of Ethics Concerns:**

The paper presents results from a human study but do not provide operationalization details. Potential risk for leading questions/biased results.

---

> ### Author Response · Authors · 2024-11-17
> **Authors’ responses to Reviewer 8txw (Part 1)**
>
> We are thankful for your work reviewing our paper. Below, we answer all of the major comments and questions, and the corresponding clarifications will be integrated into the updated version of our paper.
>
> **Q.** *“The novelty of this approach is severely lacking.”*
>
> **Response.** In our paper, we propose an innovative and effective framework for solving the email phishing attack localization problem (i.e., *this requires a method that can not only predict the vulnerability label (phishing or benign) of the email data, but also has the capability to automatically learn and figure out the most important and phishing-relevant information (i.e., sentences) of the phishing emails, providing a useful and concise explanation to users*), an area where automated machine learning and deep learning techniques have not yet been well studied.
>
> To solve the problem, our proposed AI2TALE framework starts with the selection network $\zeta$. It aims to learn and figure out the important and label-relevant information (i.e., sentences) in each email in an automatic and trainable manner. We then use information theory, specifically mutual information (as used in some baseline methods), to guide the selection process.
>
> As mentioned on page 5 of our paper, the mutual information facilitates a joint training process between the classifier $f$ and the selection process $\zeta$. The classifier learns to identify a subset of features leading to a data’s label while the selection process is designed to select the best subset of features according to the feedback of the classifier. This join training process brings benefits for selecting the important and phishing-relevant sentences from phishing emails; However, we observe two potential limitations of this training principle as follows:
>
> - Obtaining a superset of phishing-relevant sentences. The training principle of information theory does not strongly theoretically guarantee to eliminate sentences unrelated to the vulnerability of a specific email. Therefore, the set of selected sentences can be a superset of the true phishing-relevant sentences.
>
> - Encoding the vulnerability label via its selections instead of via truly meaningful information. This limitation leads to a problem in that the selected information (i.e., sentences) can be any subsets of the entire sentences and can be less likely to be meaningful ones from the data.
>
> To address these limitations, in our method, we further propose two innovative training solutions, including (i) information bottleneck theory training term and (ii) data-distribution mechanism. In particular, our innovative information bottle-neck training mechanism ensures that only important and label-relevant information (i.e., sentences) will be kept and selected while the data-distribution mechanism ensures the learnable selection process respecting the data distribution to select the meaningful and label-relevant information. It also aids in improving the generalization capability of the selection network, thereby enhancing the performance of the classifier through a random selection process that incorporates data augmentation.
>
> The extensive experiments on seven diverse real-world email datasets demonstrate the capability and effectiveness of our AI2TALE method in selecting crucial information, enabling accurate detection and offering useful and concise explanations (via the most important and phishing-relevant information triggering the classification) for the vulnerability of phishing emails. Notably, our AI2TALE method outperforms state-of-the-art baselines by 1.5% to 3.5% on average in Label-Accuracy and CognitiveTrue-Positive metrics under a weakly supervised setting, where only vulnerability labels are used without requiring ground truth phishing information.
>
> Furthermore, the qualitative results and human evaluation further demonstrate the high quality of our proposed method in solving the problem. The experimental results for the F1-score, False Positive rate, and False Negative rate further showcase the effectiveness and advancements of our AI2TALE method.

---

> ### Author Response · Authors · 2024-11-17
> **Authors’ responses to Reviewer 8txw (Part 2)**
>
> **Q.** *“This work is not the first or the only work that labels an object as phishing and identifies which features led to the classification decisions. Could the authors present a comparative analysis against existing methods, e.g., [1-2], and justify why they were not mentioned in the manuscript?”*
>
> **Response.** In our paper, we address the email phishing attack localization problem where we develop an automatic deep learning-based approach that can not only detect the vulnerability label (phishing or benign) of the email $X$ but also provide the capability to automatically learn and figure out the important and phishing-relevant information (i.e., sentences) denoted by $\tilde{X}$ (a subset of $X$) causing $X$ phishing. To the best of our knowledge, our AI2TALE method is one of the very first approaches proposed for solving the email phishing attack localization problem **(Note: We do not claim that our method is the only one, but rather that it is one of the early approaches to address this issue)**. Automated machine learning and deep learning-based techniques for this problem have not yet been well studied.
>
> *Via your comments*, you imply that there is plenty of work solving the phishing attack localization (particularly for email phishing), see e.g., [1-2] as just two random examples. We respectfully disagree, as we believe this to be an overstatement.
>
> I would like to explain how the papers [1-2] you mentioned differ and not compared in our paper, as follows:
>
> *Regarding the paper Phishpedia: A Hybrid Deep Learning-Based Approach to Visually Identify Phishing Webpages*, the authors propose a hybrid deep learning framework, Phishpedia, to address two challenges in phishing identification: (i) recognizing identity logos on webpage screenshots (using Fast R-CNN), and (ii) matching logo variants of the same brand (using a Siamese neural network model).
>
> In particular, the detected logos and URL screenshots from a webpage are compared to a corpus of target brands (collected from phishing webpages and their corresponding legitimate brands as ground truth). If the detected logos and URL screenshots match any target brands in the corpus, they are flagged as phishing; otherwise, they are considered benign. Based on this matching process, the system provides static warnings about whether the logos and URL screenshots are authentic or fake.
>
> The Phishpedia framework is specifically designed for visual data (e.g., screenshots) and is not applicable to text-based data, such as email phishing. Furthermore, the approach relies on comparing detected logos and URL screenshots to a pre-established target brand corpus, which makes it a matching-based method rather than a truly explainable AI system in the strict sense. The method only works when provided with a relevant target corpus; without it, the system cannot perform detection or explanation on its own.

---

> ### Author Response · Authors · 2024-11-17
> **Authors’ responses to Reviewer 8txw (Part 2) (continue)**
>
> *Regarding the paper An Explainable Multi-Modal Hierarchical Attention Model for Developing Phishing Threat Intelligence*, the authors propose a multi-modal hierarchical attention model (MMHAM) that jointly learns fraud cues from three major modalities of website content for phishing detection.
>
> MMHAM uses a shared dictionary learning approach to align representations from different modalities, which are learned through deep neural networks: one LSTM at the character level for character representations with character-level attention, another LSTM at the word level for word representations with word-level attention, and a ResNet for learning image representations with image-level attention. These modality-specific representations are then combined in the multi-modal hierarchical attention mechanism to create an integrated representation, which is passed through a dense layer for prediction (phishing or benign).
>
> The attention mechanisms applied at the character, word, and image levels, as used in this paper, can help highlight which elements the model focuses on, as indicated by the attention weights. However, they do not theoretically guarantee to yield truly meaningful (i.e., semantically relevant) explanations for the model's decisions. As a result, the highlighted features may appear noisy or overly broad.
>
> In contrast, using information theory, specifically through mutual information, a technique widely recognized and advanced in explainable AI, provides a more robust and interpretable solution, as demonstrated in some of our baselines. Mutual information provides a mathematical measure of the relationship between variables, quantifying how much information the input features share with the output. It helps identify which input features carry predictive information about the output variable. In short, mutual information carries some advantages over attention mechanisms used in MMHAM:
>
> *Robustness*: Mutual information is a well-established statistical measure that is less susceptible to noise and bias.
>
> *Interpretability*: Mutual information provides a clear and quantifiable measure of the relationship between variables, making it easier to understand the importance of different features.
>
> *Predictive Power*: By identifying features that are highly correlated with the output variable, mutual information can help improve the performance of the models.
>
> Additionally, the multi-modal hierarchical attention model is designed to operate at the character and word levels using LSTM, and at the image level using ResNet. Applying this model with LSTM to the sentence level (as conducted in our experiment where we found that selecting information at the sentence level provides a more complete message and clearer explanation than at the word or character level) is not straightforward and will require further modifications, including appropriate embedding and computational operations. Furthermore, the paper does not mention the release of source code for this method. We also searched GitHub for any relevant code but were unable to find it. Via your suggestion, we have referenced this paper in Section 6.5 of the revised version of our paper, where we provide details about the baselines of our proposed method.

---

> ### Author Response · Authors · 2024-11-17
> **Authors’ responses to Reviewer 8txw (Part 3)**
>
> **Q.** *”Page 3 states that post-hoc explanations do not offer a comprehensive understanding of the model's internal architecture or workings. Overexaggerated claim. Model-based XAI techniques like integrated gradients do exactly that! **Deep models are also called "self-explanatory" methods in the manuscript.** Again, popular literature dictates that deep learning is inherently a black box and thus cannot be self-explanatory/interpretable. The authors are advised to revise the manuscript and avoid redefining terms that already exist in the domain.”*
>
> **Response.** In page 3 of our paper, we wrote that:
>
> “In practice, explaining models can be divided into two categories including “post-hoc explainability techniques” and “intrinsic explainability techniques”. Post-hoc explainability techniques (e.g., LIME and SHAP) aim to elucidate the decisions of a black-box model (e.g., a deep learning model, where the internal workings are not easily understandable or interpretable) without modifying the model itself. The techniques are often applied externally to the black-box model to generate explanations specific to its predictions but do not offer a comprehensive understanding of the black-box model’s internal architecture or workings.
>
> In contrast, *intrinsic explainability techniques, a.k.a. self-explanatory models, (e.g., deep neural network-based methods with interpretable components such as L2X and AIM)* **(Note: We do not call deep models as “self-explanatory” methods)** are integrated directly into a model architecture, providing interpretability as inherent features, and offering explainability as part of their design.
>
> It is evident that intrinsic interpretable machine learning methods, a.k.a. self-explanatory models are strongly suitable for phishing attack localization because they are not only able to make predictions themselves but also have the capability to automatically learn and figure out the most important information of the data obtained from the models to explain the model’s prediction decision.”
>
> Through this discussion, we aim to highlight the distinction between these two categories of explainability techniques and emphasize the advantages and suitability of intrinsic interpretability for phishing attack localization.
>
> We also mention some key points of post-hoc explainability techniques. They are typically applied externally, providing feature-level insight into the black-box model’s decisions, without modifying the model itself. Post-hoc explainability techniques primarily focus on explaining individual predictions and do not provide an in-depth understanding of the black-box model's internal architecture, such as how the layers of a neural network are structured.
>
> Post-hoc explainability techniques are primarily used to interpret the predictions of black-box models, providing feature-level insight into the models’ decisions. These techniques are applied after the model has been trained and do not perform predictions themselves. As a result, post-hoc explanations are not suitable for solving the phishing attack localization problem.
>
> We have updated this point in the revised version of our paper to avoid any potential confusion and further improve its clarity.
>
> **Q.** *“Please add references for the datasets mentioned on page 7.”*
>
> **Response.** In our experimental evaluation, we utilized seven diverse real-world email datasets, including IWSPA-AP, Nazario Phishing Corpus, Miller Smiles Phishing Email, Phish Bowl Cornell University, Fraud Emails (Nigerian Letter), Enron Emails, and Cambridge.
>
> Six of these datasets are publicly available with associated links (i.e., Nazario: https://monkey.org/~jose/phishing/; Phish Bowl: https://it.cornell.edu/phish-bowl, Miller Smiles: http://www.millersmiles.co.uk/archives.php; IWSPA-AP: https://github.com/BarathiGanesh-HB/IWSPA-AP/tree/master/data; Enron Emails: https://www.cs.cmu.edu/~enron/; Fraud Emails: https://www.kaggle.com/datasets/rtatman/fraudulent-email-corpus) for downloading the data, while Cambridge is private. We have included citations and the associated links to the used public datasets in Section 6.2 of the revised version of our paper.
>
> **Q.** *“Please include the definition of the evaluation metrics (especially cognitive true positive) in the main body text (Section 4.2). Without the definition, it is impossible to understand the results.”*
>
> **Response.** In the first and second paragraphs on page 8 of our paper, we described the rationale, meaning, and computation of the evaluation metrics (i.e., Label-Accuracy and Cognitive-True-Positive). In the appendix on page 16, we provided additional information about Cognitive-True-Positive.

---

> ### Author Response · Authors · 2024-11-17
> **Authors’ responses to Reviewer 8txw (Part 4)**
>
> **Q.** *“The significance of the results in table 1 are hard to interpret because *the authors do not provide any information regarding the dataset*, e.g., number of phishing emails, sentences per email etc. Moreover, the improvement in the results is marginal (1-3%). Does the proposed approach provide benefits beyond the results in Table 1?”*
>
> **Response.** On page 7 of our paper, we briefly described the seven datasets studied. *Detailed information about the number of samples and characteristics of the datasets is provided in the appendix on page 15*. Based on your suggestion, we will incorporate this information into the main text in the updated version of our paper.
>
> For the quantitative results, under the weakly supervised setting, when dealing with complex data (e.g., emails written from various writing styles, structures, and sources) and only the most important (top-1) selected sentences from emails are utilized, the observed improvement of our AI2TALE method from around 1.5% to 3.5% in the combined average performance of the Label-Accuracy and Cognitive-True-Positive measures, especially within the range over 99% and approaching 100%, signifies a substantial advancement.
>
> The qualitative results (shown in Table 2) and human evaluation (presented with Figure 2) further demonstrate the quality of our AI2TALE method in solving the email phishing attack localization problem (i.e., not only can detect the vulnerability label (phishing or benign) of the email data but also can automatically learn and identify the most important and phishing-relevant information (e.g., sentences) in phishing emails).
>
> **Q.** *“Please also report false positives and false negatives for each method in table 1. FPs are especially a big problem in cybersecurity tasks.”*
>
> **Response.** In the first paragraph on page 10 of our paper, we reported the false positive rate and false negative rate of our AI2TALE method. Specifically, our method achieves a False Positive rate of 0.451% and a False Negative rate of 0.899%.
>
> In addition, in the appendix on page 18, we computed and reported the F1-score in two scenarios: (i) exclusively on the testing phishing emails, and (ii) on both the testing phishing and benign emails of our AI2TALE method and the baselines (i.e., AIM, VIBI, L2X, ICVH, and INVASIVE).  The F1-score is the harmonic mean of Precision and Recall. Therefore, a high F1-score suggests that the model is minimizing both false positives and false negatives. As shown in Table 3, all baselines have an F1-score from and greater than 96.63%, while our AI2TALE method achieves an F1-score of 99.33%.
>
> Based on your suggestion, we have computed the False Positive rate and False Negative rate for the baselines and have presented these values alongside the F1-score in Section 6.7 of the revised version of our paper.
>
> The False Positive rate (FPR) and False Negative rate (FNR) for our AI2TALE method and the baselines are summarized as follows:
>
> INVASE (FPR: 2.353% and FNR: 1.048% )
>
> ICVH  (FPR: 3.355% and FNR: 3.195% )
>
> L2X (FPR: 2.253% and FNR: 1.248% )
>
> VIBI (FPR: 2.504% and FNR: 4.194% )
>
> AIM (FPR: 1.853% and FNR: 1.348% )
>
> AI2TALE (FPR: 0.451% and FNR: 0.899% )

---

> ### Author Response · Authors · 2024-11-17
> **Authors’ responses to Reviewer 8txw (Part 5)**
>
> **Q.** *“Please provide more details regarding the human study. 25 participants don't seem statistically significant for 10 emails, each with 5 options, as estimated by Cochran's formula. Moreover, it is unclear what questions are asked or how they are phrased exactly. For instance, if the participants are always shown the top-1 sentence and not given an option of top-3 sentences, this can very likely be interpreted as a rhetorical question, which leads people to respond in the affirmative most of the time.”*
>
> **Response.** Human evaluation is an additional measure used to assess the quality of our AI2TALE method for email phishing attack localization. In our human evaluation, there were 25 experienced participants, including university students and staff (i.e., lecturers, professors, engineers, research scientists, and research fellows), representing diverse professional backgrounds, education levels, career stages, and age groups. All participants responded that they have used emails for work and study, and they have both experienced and heard about phishing attacks.
>
> Each participant is asked to evaluate the selected sentences of 10 different phishing emails (randomly chosen from the testing set) in terms of whether the sentence selected in each email is important to influence and persuade users to follow the instructions in the emails. The selected sentence was presented alongside all other sentences from each email, allowing participants to compare it with the others and provide a more objective and quantified assessment.
>
> *An example question:
>
> Giving an email as follows.  Do you think the selected sentence (in bold) affects and persuades users' decision to follow the instructions from the email?
>
> “Bulk attention! Your discover account will close soon! **Dear member, we have faced some problems with your account, so please update the account.** If you do not update will be closed. To update your account, just confirm your information. (it only takes a minute). It's easy. 1. Click the link below to open a secure browser window. 2. Confirm that you're the owner of the account, and then follow the instructions."
>
> The participants will then choose one option from the following: Strongly Agree, Agree, Neutral, Disagree, or Strongly Disagree.*
>
> Furthermore, as mentioned in the second paragraph on page 10 of our paper, in this human evaluation, we implemented careful study design protocols to minimize potential priming. Particularly, to ensure objectivity in the results, no information was provided about the source of the selected sentences.
>
> We have updated this information in the Human Evaluation section and in Section 6.12 of the revised version of our paper.
>
> **Q.** *“Overall, the terminology used in this paper, e.g., phishing attack localization and phishing vulnerability are atypical for the cybersecurity community. Page 1, para 2 states that AI has been used for malware vulnerability detection. Malware detection and vulnerable code detection are two entirely different sub-domains.”*
>
> **Response.** On page 1, paragraph 2, we briefly present the success and benefits of using artificial intelligence in various domains such as autonomous driving, data generation, drug discovery, and malware and software vulnerability detection. By leveraging the power of machine learning and deep learning, there have also been many efforts proposed for solving phishing attack problems. We have updated this paragraph to make it clearer in the revised version of our paper.

---

> > ### Comment · Reviewer_8txw · 2024-11-22
> >
> > I thank the authors for taking the time to provide a thorough response. However, the weaknesses I point out are not sufficiently addressed:
> > - Novelty: Text-based phishing detection works have long preceded the visual-based detection works. Thus, there is a significant body of literature that has not even been mentioned in the manuscript, let alone compared against, in my opinion. The original comment was to rephrase the seemingly overexaggerated claims about novelty and compare against appropriate baselines.
> > - Explainability: Generally speaking, techniques do not just become explainable just because they can be visualized or are novel. There is insufficient motivation in the text for why MI is considered interpretable, and why attention mechanism is not compared against. Please provide a more detailed and concrete reasoning for this.
> > - Performance: An improvement delta of ~3% is marginal at best, and for this approach to be considered significant, it must provide added utility from other aspects (especially from a cybersecurity perspective), which I find lacking in this work, especially considering that related work has not been covered sufficiently well.
> >
> > Based on the response, I am inclined to keep my current rating.

---

> > > ### Author Response · Authors · 2024-11-23
> > > **Authors’ responses to Reviewer 8txw (Part 6)**
> > >
> > > We thank you for your feedback. We would like to answer your remaining questions and concerns as follows:
> > >
> > > **Q.** *"Novelty: Text-based phishing detection works have long preceded the visual-based detection works. Thus, there is a significant body of literature that has not even been mentioned in the manuscript, let alone compared against, in my opinion. The original comment was to rephrase the seemingly overexaggerated claims about novelty and compare against appropriate baselines."*
> > >
> > > **Response.** As mentioned on page 1 of our paper and in previous responses (Part 1), in our paper, we study the phishing attack localization problem, specifically in the context of email phishing, that goes beyond the email phishing detection where the corresponding models can only predict the label (phishing or benign) of the email and lack of the capability in providing explanations that offer concise and meaningful interpretations of the information causing the data phishing. *In our paper, we cited numerous phishing detection methods and highlighted this limitation.*
> > >
> > > The phishing attack localization problem requires a method that not only predicts the vulnerability label (phishing or benign) of the emails, but also automatically identifies and learns the most important and phishing-relevant information (e.g., key sentences) of the phishing emails, providing concise and useful explanations to users. To the best of our knowledge, automated machine learning and deep learning techniques for this problem have not yet been well-studied. *For details on the novelty of our approach and how our method addresses this problem, please refer to our previous response (Part 1) or Section 3.2 of our paper.*
> > >
> > > In our paper, *we comprehensively presented the related baselines capable of solving the problem*. In our experiments, we compared our method with several recent, popular, and state-of-the-art intrinsic interpretability methods, a.k.a. self-explanatory models. These models can make predictions and have the capability to identify the most important features of the data to explain the model's decisions in the context of email phishing attack localization.
> > >
> > > In previous responses (Part 2), we have explained in detail why the two papers you mentioned, including (1) "Phishpedia: A Hybrid Deep Learning-Based Approach to Visually Identify Phishing Webpages" and (2) "An Explainable Multi-Modal Hierarchical Attention Model for Developing Phishing Threat Intelligence", are different and were not compared in our paper.
> > >
> > > **Q.** *"Performance: An improvement delta of ~3% is marginal at best, and for this approach to be considered significant, it must provide added utility from other aspects (especially from a cybersecurity perspective), which I find lacking in this work, especially considering that related work has not been covered sufficiently well."*
> > >
> > > **Response.** Our study addresses the phishing attack localization problem, specifically in the context of email phishing, where we propose a method that can not only predict the vulnerability label (phishing or benign) of email data, but also has the intrinsic ability to automatically identify the most important and phishing-relevant information (i.e., sentences) in phishing emails, providing a useful and concise explanation. In our paper, we have referenced a comprehensive set of related baselines.
> > >
> > > Our experiments are conducted in *a weakly supervised setting*, where only vulnerability labels (phishing or benign) are available, and the ground truth phishing information is absent. The experiments are performed on *seven diverse real-world email datasets*, using the most important (top-1) selected sentences from the emails to provide the most highly qualified and concise explanation to users.
> > >
> > > Given this experimental setup, our AI2TALE method outperforms state-of-the-art baselines by 1.5% to 3.5% on average in the Label-Accuracy and Cognitive-True-Positive metrics. Notably, the baselines already achieve very high performance (e.g., many surpass 98% in Label-Accuracy, and most exceed 97%, with the second-highest baseline also above 98% in Cognitive-True-Positive). The improvement achieved by our method, therefore, signifies a substantial advancement.
> > >
> > > Moreover, the extensive qualitative results and human evaluation further demonstrate the high quality of our AI2TALE method in solving the problem. The experimental results for the F1-score, False Positive rate, and False Negative rate also highlight the effectiveness and advancements of our AI2TALE method.

---

> > > > ### Author Response · Authors · 2024-11-23
> > > > **Authors’ responses to Reviewer 8txw (Part 7)**
> > > >
> > > > **Q.** *"Explainability: Generally speaking, techniques do not just become explainable just because they can be visualized or are novel. There is insufficient motivation in the text for why MI is considered interpretable, and why attention mechanism is not compared against. Please provide a more detailed and concrete reasoning for this."*
> > > >
> > > > **Response.** Explainable machine learning is an important research area focused on developing methods that help to understand and interpret the decisions made by machine learning models. These techniques (especially intrinsic explainability techniques, a.k.a. self-explanatory models) aim to make the inner workings of complex models more transparent, enabling users to trust and validate the predictions made by the system.
> > > >
> > > > Our method is a self-explanatory model designed to solve the phishing attack localization problem, particularly for email phishing. In particular, our method can not only detect the vulnerability label (phishing or benign) of the email $X$ but also has the capability to automatically learn and figure out the important and phishing-relevant information (i.e., sentences) denoted by $\tilde{X}$ (a subset of $X$) causing $X$ phishing. *Refer to Section 3.2 in our paper or our previous responses (Part 1) to have details about how our method works and addresses the email phishing attack localization problem.*
> > > >
> > > > Using information theory, specifically through mutual information (MI), a technique widely recognized and advanced in explainable AI, provides a robust and interpretable solution, as also demonstrated by some of our baselines (e.g., L2X, VIBI, and AIM). Mutual information provides a mathematical measure of the relationship between variables, quantifying how much information the input features share with the output. This helps identify which input features carry predictive information about the output variable.
> > > >
> > > > *Regarding the attention mechanism you mentioned*, which is discussed in the paper “An Explainable Multi-Modal Hierarchical Attention Model for Developing Phishing Threat Intelligence”, we have already provided a detailed explanation in our previous response (Part 2) on why this method is not compared in our paper. We would like to summarise it as follows:
> > > >
> > > > - This method is designed for website phishing detection, utilizing attention mechanisms applied at the character, word, and image levels. These mechanisms highlight the elements the model focuses on, as indicated by the attention weights. However, in general, they do not theoretically guarantee truly meaningful (i.e., semantically relevant) explanations for the model's decisions. In contrast, the use of information theory, specifically mutual information, in explainable AI is grounded in a solid theoretical foundation.
> > > >
> > > > - Additionally, this method is designed to operate at the character and word levels using LSTM, and at the image level using ResNet. Applying this model using attention mechanism with LSTM to the sentence level (*as conducted in our experiment where we found that selecting information at the sentence level provides a more complete message and clearer explanation than at the word or character level*) is not straightforward and will require further modifications, including appropriate embedding and computational operations. Furthermore, the paper does not mention the release of source code for this method. We also searched GitHub for any relevant code but were unable to find it.
> > > >
> > > > Due to these limitations, we do not use this method as our baseline for solving the email phishing attack localization problem in our paper.
> > > >
> > > > In response to your request, we have attempted to implement this model by using the attention mechanism with LSTM to address the email phishing attack localization problem. To adapt this approach to work at the sentence level, we used the same sentence embedding technique as in our proposed method. In this setup, the attention weights reflect the importance of each sentence in the model's prediction. During the training process, we varied the LSTM hidden size (a hyperparameter of this model) across {8, 16, 32, 64} as suggested in this paper. As a result, this model achieves a Label-Accuracy of up to 98.55% and a Cognitive-True-Positive of up to 95.31%. *Our method outperforms this model by 2.21% on average of the Label-Accuracy and Cognitive-True-Positive metrics.*
> > > >
> > > > Moreover, we observe that the performance of this model is highly sensitive to changes in its hyperparameters. Specifically, its performance on the Cognitive-True-Positive metric fluctuates between 82.73% and 95.31%. In contrast, our method exhibits significantly greater stability, with much smaller variations in response to hyperparameter changes (see our responses to Reviewer YZX3, Part 3, for details). In summary, our AI2TALE method achieves 98.95% on the Cognitive-True-Positive metric with a variance of just 0.0852, highlighting the robustness of our approach.

---

> > > > > ### Comment · Reviewer_8txw · 2024-11-25
> > > > >
> > > > > I greatly appreciate the authors' thorough responses. I do have to say that most of the response text is either a repetition of previous responses or text from the paper, which I have already read, so unfortunately it does little to convince me. The two papers I mentioned were random examples (as stated in original review as well) to show that there is plenty of work in the phishing detection realm that the authors do not cover. Now whether it uses a website or an email service as a base has no consequence since both operate (somewhat) on text modalities. Moreover, the technical terminology is also being reinvented in the paper, which may be a reason why the authors could not find existing work in this domain. The attention mechanism, for instance, is not specific to that one paper but rather a general technique that many works within and outside cybersecurity have used for explaining neural networks. The performance improvement of 2% is marginal still and there is no explanation for _why_ this proposed method should be used over other already existing/well-established methods. Therefore, I am still inclined to keep my score.

---

> ### Author Response · Authors · 2024-11-26
> **Authors’ responses to Reviewer 8txw (Part 8)**
>
> We thank you for reviewing our responses.
>
> In our previous replies, we summarized and reiterated key points from both our paper and earlier responses, as we believe some important information we provided (about the scope of our study, the novelty, and the advancements and effectiveness not only through quantitative experiments across a wide range of metrics (Label-Accuracy, Cognitive-True-Positive, F1-score, False Positive Rate, and False Negative Rate) on seven diverse real-world datasets but also through qualitative and human evaluations) may have been overlooked.
>
> In this response, we would like to address your additional comments to further clarify the novelty, effectiveness, and advancements of our work in solving the email phishing attack localization problem.
>
> **Q**. *”The technical terminology is also being reinvented in the paper, which may be a reason why the authors could not find existing work in this domain.”*
>
> The distinction between phishing detection and phishing localization is not merely a matter of technical terminology (as mentioned in your comments). It reflects two fundamentally different challenges: phishing detection involves classifying emails as phishing or benign, while *phishing localization (a more difficult task)*, as mentioned in our paper, goes beyond detection by focusing on identifying and understanding the specific content (e.g., sentences) within an email that triggers the phishing classification.
>
> In our paper, we provided a thorough review of relevant baselines capable of solving the problem and compared our method with a range of recent, widely used, and state-of-the-art approaches.
>
> **Q.** *"The attention mechanism."*
>
> We acknowledge that the attention mechanism can help explain what parts of the input the model is focusing on when making a decision. *However, it does not theoretically guarantee to yield truly meaningful (i.e., semantically relevant) explanations for the model's decisions*.  In this regard, mutual information (as used in our method and some of the baselines) is theoretically more grounded than the attention mechanism, especially due to its information-theoretic foundations, in providing a more robust and interpretable solution. In particular, mutual information can be used to quantify how much information a feature (or a subset of features) provides about the output (or class label). This is grounded in solid theory and has been studied for many years in areas like statistical learning, feature selection, and information theory.
>
> With regard to the work (cited in your comments) that uses the attention mechanism (i.e., this work combines the attention mechanism with LSTM networks, a common and widely adopted approach in various natural language processing tasks), in our responses, we have adapted this method to address the problem. Both the experimental results and the stability of our approach demonstrate significant advancements and robustness compared to this method. *Notably, the high sensitivity of this approach to changes in its hyperparameters, particularly on the Cognitive-True-Positive metric, highlights the limitation in the robustness of the attention mechanism in obtaining meaningful features aligned with human reasoning. In contrast, our method achieves much higher performance on the Cognitive-True-Positive metric with significantly smaller variance.*
>
> **Q.** *"There is no explanation for why this proposed method should be used over other already existing/well-established methods.”*
>
> In the framework section (Section 3.2) of our paper and our previous responses (Part 2), we clearly explained the novelty of our method in solving the email phishing attack localization problem. Our proposed solution is theoretically guaranteed to select important, meaningful, label-relevant information for explainability, going beyond mere detection.
>
> In our paper, we comprehensively presented relevant baselines capable of solving the problem and compared our method with many recent, popular, and state-of-the-art approaches. The extensive experimental results, conducted on seven diverse real-world email datasets, *demonstrate the effectiveness and advancements of our method*, both through quantitative experiments (across a wide range of metrics, including Label-Accuracy, Cognitive-True-Positive, F1-score, False Positive Rate, and False Negative Rate) and through qualitative and human evaluations.
>
> In addition, the improvements and robustness of our method, compared to the approach that combines the attention mechanism with LSTM networks (a solution that can be adapted to solve the email phishing attack localization problem, as cited in your comments), further highlight the contribution of our work.

---

### Official Review · Reviewer_PCAG · 2024-11-02

**Soundness:** 4
**Presentation:** 4
**Contribution:** 3
**Rating:** 8
**Confidence:** 3

**Summary:**

This work addresses the problem of phishing attacks and introduces an information theory-based model called AI2TALE to detect them without requiring ground truth while also providing explainable results. The authors validate this model on seven diverse real-world email datasets, demonstrating that the AI2TALE model achieves state-of-the-art results. The model is also tested with human participants, who found it helpful for detecting phishing attacks.

**Strengths:**

Overall, this is a well-written and clearly explained work. The motivation and contributions are effectively communicated, as are the details of the algorithm, including the model architecture and data. According to the results presented in the tables, the AI2TALE model achieves state-of-the-art performance in detecting phishing attacks based on two measures—Label-Accuracy and Cognitive-True-Positive—which the authors believe are more appropriate for this task. The model also achieves state-of-the-art results with the well-known F1 score. The results are also validated by humans who found this model helpful. Another strength of this paper is that the results of this information-theory-based model are explainable, a claim that appears valid based on the presented figures. Finally, all code is reproducible and open-source.

**Weaknesses:**

I have included a few comments regarding the presentation and also have some feedback on the experimental section.

   A) Presentation Comments

   A1) In lines 058 to 060, where you mention that early-stage AI approaches are among the most effective solutions for preventing and reducing negative effects, this seems like a strong statement. It would be beneficial to include a citation to support this claim.

   A2) In line 152, you introduce the term "AI2TALE" for the first time as the name of your model. It would be helpful to clarify what this acronym stands for and briefly explain your rationale behind the name selection.

   A3) In line 330, you refer to your method as “Algorithm 1.” Since this is the only algorithm presented in your paper, it is unnecessary to number it as "1." Consider renaming it simply as “Algorithm” throughout the text.

   A4) In Section 4.1, “Studied Datasets” (lines 360-371), you list several datasets used in your research. Please add citations and links for each referenced dataset. Additionally, include citations and links in Section 6.2 of the appendices (lines 778-788).

   A5) In line 404, you mention that readers seeking more details can refer to the appendices. Please specify the exact section or appendix that contains the relevant information.

   A6) In Section 6.2 of the appendices (lines 789-799), this paragraph appears to be more relevant to the following section, 6.3, on data preprocessing and embeddings.

   A7) The link of your code, see line 897, should be in the main paper, such as in the introduction.

A8) As mentioned in the author guidelines (see https://iclr.cc/Conferences/2025/AuthorGuide), you are encouraged to include a Reproducibility Statement at the end of the main text. This statement should detail the efforts made to ensure reproducibility and include any necessary information for those wishing to replicate your results.

   A9) You have referenced a good selection of papers with nice variation; however, I think a few relevant papers are missing. These include "Feature-based Learning for Diverse and Privacy-Preserving Counterfactual Explanations" by Vy Vo et al., "The Anatomy of Deception: Measuring Technical and Human Factors of a Large-Scale Phishing Campaign" by Anargyros Chrysanthou et al., "Towards Modeling Uncertainties of Self-Explaining Neural Networks via Conformal Prediction" by Wei Qian et al., and "DIB-X: Formulating Explainability Principles for a Self-Explainable Model Through Information Theoretic Learning" by Changkyu Choi et al. In the related work section of the appendices or the main paper, you could also include additional studies based on information theory that are not necessarily related to phishing attacks.

B) Experimental Section Comments

   B1) In Table 1 (line 424), you present the results for each model. Since you used multiple datasets, it would be helpful to show the results for each dataset separately.

   B2) In Table 2 (line 443), it would be nice to include the second and third sentence top sentences of the model highlighting them with different colors and briefly discussing them.

   B3) In the human evaluation section (line 493), please include any available statistics on the participants, such as their expertise, gender, education level, etc.

   B4) In Section 6.7 of the appendices (lines 899–960), Table 3 presents results that, in my opinion, are quite significant and provide further support for the strength of your model. It would be better to incorporate the results from Table 3 into Table 1 and remove the "average" section.

**Questions:**

1) In line 271, you mention that $p(\tilde{x_i}|X)$ is a Gaussian mixture distribution. Why did you choose this distribution? Did you evaluate the effect of this assumption on your results?

  2) In your algorithm (see line 333), you split an email into sentences based on periods or commas. Have you examined whether this sentence-splitting method impacts your results? There are alternative ways to split text, such as by a specified number of characters.

  3) Based on the main paper and appendices, it appears you developed a feed-forward neural network (see lines 879–881) for the selection model in your algorithm (see line 339). My understanding is that this neural network creates word embeddings based on the provided texts and then focuses on detecting phishing attacks. Why did you choose this model instead of a more advanced architecture, such as a transformer?

**Details Of Ethics Concerns:**

To the best of my knowledge, I see no ethical concerns regarding this work.

---

> ### Author Response · Authors · 2024-11-17
> **Authors’ responses to Reviewer PCAG (Part 1)**
>
> We appreciate the effort you have dedicated to reviewing our paper and offering insightful suggestions for further enhancements and clarifications. Your recognition of our work is greatly appreciated. Below, we answer all of the major comments and questions, and the corresponding clarifications will be integrated into the updated version of our paper.
>
> **Q.** *“In line 271, you mention that $p(\tilde{\boldsymbol{x}\_{i}}|X)$ is a Gaussian mixture distribution. Why did you choose this distribution? Did you evaluate the effect of this assumption on your results?”*
>
> **Response.** The objective function in formula (4) aims to guide the selection process so that only important and label-relevant information (i.e., sentences) is selected. This is achieved by minimizing the mutual information between $\tilde{{X}}$ and $X$ while maximizing the mutual information between $\tilde{X}$ and $Y$.
>
> To minimize the mutual information between $X$ and $\tilde{{X}}$, we derive the objective functions in formulas (5) and (6). As a result, minimizing $\mathbf{I}(X,\tilde{X})$ is equivalent to minimizing the KL divergence between $p(\tilde{\boldsymbol{x}\_{i}}|X)$ and $r(\tilde{\boldsymbol{x}\_{i}})$ with $i$ from $1$ to $L$. Derived from the formulas (4,5,6), one can view $r(\tilde{\boldsymbol{x}\_{i}})$ as the prior distribution which is constructed by $r(\tilde{\boldsymbol{x}\_{i}})=\mathcal{N}(\tilde{\boldsymbol{x}}\_{i}|0,\sigma^{2})$. Then, the posterior $p(\tilde{\boldsymbol{x}}\_{i}|X)$ is a Gaussian mixture distribution (i.e., between $p\_{i}\mathcal{N}(\tilde{\boldsymbol{x}}\_{i}|\boldsymbol{x}\_{i},\sigma^{2})$ and ($1-p\_{i})\mathcal{N}(\tilde{\boldsymbol{x}\_{i}}|0,\sigma^{2})$ where $\sigma>0$ is a small number), the intuition is that the prior prefers the small values centered at $0$. In this way, $p(\tilde{\boldsymbol{x}}\_{i}|X)$ is encouraged to select fewer sentences. $D_{KL}(p(\tilde{\boldsymbol{x}}\_{i}|X)\Vert r(\tilde{\boldsymbol{x}}\_{i}))$ can be computed by the approximation objective function in formula (7).
>
> The assumption of a Gaussian prior has been shown to work well in practice across a wide range of domains and applications (Bishop, 2006). When no strong prior knowledge is available, a Gaussian prior is a reasonable choice, both theoretically grounded (via the Central Limit Theorem) and empirically effective across various fields (Gelman et al., 2013; Hogg et al., 2019).
>
> A Gaussian mixture distribution is versatile, capable of approximating a wide range of probability distributions and capturing relationships both within the data and between variables (Bishop, 2006). Moreover, when $p(\tilde{\boldsymbol{x}\_{i}}|X)$ is the Gaussian mixture distribution, we also gain the following benefit:
>
> - It facilitates the minimization of the KL divergence between $p(\tilde{\boldsymbol{x}\_{i}}|X)$ and $r(\tilde{\boldsymbol{x}\_{i}})$. The KL divergence between a Gaussian mixture and a Gaussian prior is well-defined and tractable, enabling us to effectively regularize the model and improve interpretability. The Gaussian mixture's components allow us to explicitly control the balance between fidelity to the data ($p(\tilde{\boldsymbol{x}\_{i}}|X)$) and the prior belief about the distribution of $\tilde{\boldsymbol{x}\_{i}}$ (as defined by $r(\tilde{\boldsymbol{x}\_{i}})$).
>
> The effectiveness and advancements of our AI2TALE method for solving the phishing attack localization problem (not only predict the vulnerability label (i.e., phishing or benign) of the email data, but also automatically learn and figure out the most important and phishing-relevant information (i.e., sentences) triggering the classification) are demonstrated through extensive experiments on seven diverse real-world email datasets. The best performance in terms of Label-Accuracy, Cognitive-True-Positive, F1-score, False Positive rate, and False Negative rate compared to the state-of-the-art baselines, underscore the strengths of both the chosen prior distribution and the overall approach.
>
> References:
>
> Christopher M. Bishop. Pattern Recognition and Machine Learning. Springer. 2006.
>
> Andrew Gelman, John B. Carlin, Hal S. Stern, David B. Dunson, Aki Vehtari, and Donald B. Rubin. Bayesian Data Analysis. Chapman and Hall. 2013.
>
> Robert Hogg, Joseph McKean, and Allen Craig. Introduction to Mathematical Statistics. Pearson. 2019.

---

> > ### Comment · Reviewer_PCAG · 2024-11-25
> >
> > I would like to thank the authors for taking the time to answer my questions and provide further details. Based on your responses, I am satisfied with most of the points I raised. However, there is still one question (see below) that I would appreciate if you could elaborate on further:
> >
> > Q. “Based on the main paper and appendices, it appears you developed a feed-forward neural network (see lines 879–881) for the selection model in your algorithm (see line 339). My understanding is that this neural network creates word embeddings based on the provided texts and then focuses on detecting phishing attacks. Why did you choose this model instead of a more advanced architecture, such as a transformer?”
> >
> > I understand your comment regarding the substantial computational resources required for a transformer model, as well as the additional modifications needed to adapt it to your task. However, this alone does not explain why you did not compare your approach with a state-of-the-art model. I acknowledge that training such a model from scratch can be time-consuming, but have you considered leveraging pre-trained word embeddings or testing a pre-trained transformer model by fine-tuning it for your specific task?
> >
> > Additionally, based on the comments of reviewer 8txw, who raised a similarly useful point, I would suggest that further work is needed in this area.

---

> ### Author Response · Authors · 2024-11-17
> **Authors’ responses to Reviewer PCAG (Part 2)**
>
> **Q.** *“In your algorithm (see line 333), you split an email into sentences based on periods or commas. Have you examined whether this sentence-splitting method impacts your results? There are alternative ways to split text, such as by a specified number of characters.”*
>
> **Response.** In our study, we view each email as a sequence of sentences and aim to identify the most important sentence contributing to the email's vulnerability label (phishing or benign). Through our observations, we found that selecting information at the sentence level provides a more complete message and clearer explanation than focusing on keywords. Therefore, we decided to conduct our experiments at the sentence level.
>
> To split each email into sentences, we use the Natural Language Toolkit (NLTK), a widely used tool for sentence tokenization. NLTK employs a set of rules and algorithms that go beyond simple period-based segmentation, allowing it to accurately identify sentence boundaries. While alternative methods, such as splitting by a specified number of characters, may yield different results, we chose NLTK for its simplicity, effectiveness, and wide adoption in text processing tasks.
>
> **Q.** *“Based on the main paper and appendices, it appears you developed a feed-forward neural network (see lines 879–881) for the selection model in your algorithm (see line 339). My understanding is that this neural network creates word embeddings based on the provided texts and then focuses on detecting phishing attacks. Why did you choose this model instead of a more advanced architecture, such as a transformer?”*
>
> **Response.** As mentioned on page 15 in the appendix of our paper, to embed each sentence (a sequence of tokens, i.e., we also apply NLTK for word tokenization) into a vector, we drew inspiration from the baselines. In particular, we used a 150-dimensional token Embedding layer followed by a Dropout layer with a dropped fixed probability $p=0.2$, a 1D convolutional layer with the filter size $150$ and kernel size $3$, and a 1D max pooling layer. Finally, a mini-batch of emails in which each email consisting of $L$ encoded sentences was fed to our proposed AI2TALE method and the baselines. It is worth noting that the Embedding and 1D convolutional layers are learnable during the training process. In our method, the selection network receives the encoded sentences and is primarily responsible for selecting label-relevant features through the joint training process with the classifier.
>
> The combination of the Embedding layer, 1D convolutional layer, and 1D max pooling layer is known to be a lightweight (resource-efficient) and effective approach for processing text data and learning representations. This method has the ability to capture patterns and logical relationships in text through convolutional and pooling layers with non-linear activations.
>
> Using a transformer's architecture is another method for data embedding and representation. However, it requires substantial computing resources. Furthermore, the transformer's architecture is primarily designed to operate at the token level. To work at the sentence level, it requires extra layers or architectural modifications, introducing additional complexity, making it more difficult to train and fine-tune.
>
> **Q.** *“In lines 058 to 060, where you mention that early-stage AI approaches are among the most effective solutions for preventing and reducing negative effects, this seems like a strong statement. It would be beneficial to include a citation to support this claim.”*
>
> **Response.** Thank you for your helpful comment. The statement could be updated to: “It has been proven that utilizing AI-based approaches (i.e., machine learning and deep learning-based algorithms) to detect phishing attacks in the early stages is one of effective solutions for preventing and reducing the negative effects caused”. We will include some references to support this claim, including:
>
> Sultan Asiri, Yang Xiao, Saleh Alzahrani, and Tieshan Li. PhishingRTDS: A real-time detection system for phishing attacks using a Deep Learning model. Computers and Security. 2024.
>
> Bilal Naqvi, Kseniia Perova, Ali Farooq, Imran Makhdoom, Shola Oyedeji, and Jari Porras. Mitigation strategies against the phishing attacks: A systematic literature review. Computers and Security. 2023.
>
> Abdul Basit, Maham Zafar, Xuan Liu, Abdul Rehman Javed, Zunera Jalil, Kashif Kifayat. A comprehensive survey of AI-enabled phishing attacks detection techniques. Telecommunication Systems. 2020.
>
> We have updated this point in Section 1 of the revised version of our paper.

---

> ### Author Response · Authors · 2024-11-17
> **Authors’ responses to Reviewer PCAG (Part 3)**
>
> **Q.** *“In Section 4.1, “Studied Datasets” (lines 360-371), you list several datasets used in your research. Please add citations and links for each referenced dataset. Additionally, include citations and links in Section 6.2 of the appendices (lines 778-788).”*
>
> **Response.** In our experimental evaluation, we utilized seven diverse real-world email datasets, including IWSPA-AP, Nazario Phishing Corpus, Miller Smiles Phishing Email, Phish Bowl Cornell University, Fraud Emails (Nigerian Letter), Enron Emails, and Cambridge.
>
> Six of these datasets are publicly available with associated links (i.e., Nazario: https://monkey.org/~jose/phishing/; Phish Bowl: https://it.cornell.edu/phish-bowl, Miller Smiles: http://www.millersmiles.co.uk/archives.php; IWSPA-AP: https://github.com/BarathiGanesh-HB/IWSPA-AP/tree/master/data; Enron Emails: https://www.cs.cmu.edu/~enron/; Fraud Emails: https://www.kaggle.com/datasets/rtatman/fraudulent-email-corpus) for downloading the data, while Cambridge is private. We have included citations and the associated links to the used public datasets in Section 6.2 of the revised version of our paper.
>
> **Q.** *“In line 404, you mention that readers seeking more details can refer to the appendices. Please specify the exact section or appendix that contains the relevant information.”*
>
> **Response.** Thank you for pointing this out. We have fixed this in the revised version of our paper.
>
> **Q.** *“You have referenced a good selection of papers with nice variation; however, I think a few relevant papers are missing. These include "Feature-based Learning for Diverse and Privacy-Preserving Counterfactual Explanations" by Vy Vo et al., "The Anatomy of Deception: Measuring Technical and Human Factors of a Large-Scale Phishing Campaign" by Anargyros Chrysanthou et al., "Towards Modeling Uncertainties of Self-Explaining Neural Networks via Conformal Prediction" by Wei Qian et al., and "DIB-X: Formulating Explainability Principles for a Self-Explainable Model Through Information Theoretic Learning" by Changkyu Choi et al. In the related work section of the appendices or the main paper, you could also include additional studies based on information theory that are not necessarily related to phishing attacks.”*
>
> **Response.** Thank you for suggesting these interesting papers. We have updated the Related Work section to include these references.

---

> ### Author Response · Authors · 2024-11-17
> **Authors’ responses to Reviewer PCAG (Part 4)**
>
> **Q.** *“In Table 2 (line 443), it would be nice to include the second and third sentence top sentences of the model highlighting them with different colors and briefly discussing them.”*
>
> **Response.** Thank you for your insightful suggestion.
>
> In our paper, with the aim of providing the most highly qualified and concise explanation of the vulnerability of email data to users, we primarily assess the model’s performance based on the most important (top-1) selected sentence from each email.
>
> We agree that including the second and third most relevant sentences would enhance the utility for users. When deploying our AI2TALE method in real-world applications, offering the option to display the top-2 or top-3 sentences alongside the top-1 sentence could be beneficial.
>
> We have conducted this additional analysis, and the results are presented in Section 6.8 of the revised version of our paper, complementing those shown in Table 2. Below, we present an example of the top-3 selected sentences from the email shown in Table 2. This qualitative result further demonstrates the effectiveness and advantages of our proposed AI2TALE method for solving the email phishing attack localization problem. The selected sentences support each other in explaining the predicted label.
>
> Email: “Bulk attention! Your discover account will close soon! Dear member, we have faced some problems with your account, so please update the account. If you do not update will be closed. To update your account, just confirm your information. (it only takes a minute). It's easy. 1. Click the link below to open a secure browser window. 2. Confirm that you're the owner of the account, and then follow the instructions."
>
> Selected sentences:
>
> 1st: Dear member, we have faced some problems with your account, so please update the account.
>
> 2nd: To update your account, just confirm your information.
>
> 3rd: It's easy.
>
> The message from the first selected sentence obtained from our method exhibits cognitive triggers commonly associated with phishing attempts used in the phishing email. In particular, it implies a sense of urgency (concern) via problems with your account while “Dear member" aims to establish a connection with the recipient and imply that the message comes from a trusted source. The phrase “please update the account" creates a sense of familiarity and consistency.
>
> While the statement “It's easy” and the instruction to “just confirm your information” (via the second and third selected sentences) aim to minimize perceived effort, making the recipient more likely to comply without hesitation.
>
> We believe these additional results will further demonstrate the effectiveness and advancement of our AI2TALE method in solving the email phishing attack localization problem.
>
> **Q.** *“In the human evaluation section (line 493), please include any available statistics on the participants, such as their expertise, gender, education level, etc.”*
>
> **Response.** In our paper, human evaluation serves as an additional measure to assess the quality of our AI2TALE method for email phishing attack localization. The evaluation involved 25 experienced participants, *including university students and staff (i.e., lecturers, professors, engineers, research scientists, and research fellows), representing diverse professional backgrounds, education levels, career stages, and age groups*. In response to your suggestion, as well as feedback from other reviewers, we have updated this information in the Human Evaluation section of the revised version of our paper.
>
> **Qs and Responses.** Regarding the other comments on the presentation and experiments, following your suggestions, we have made the necessary modifications and updates to further enhance these aspects of our paper.

---

> ### Author Response · Authors · 2024-11-25
> **Authors’ responses to Reviewer PCAG (Part 5)**
>
> We thank you for your feedback. We would like to address your remaining question as follows:
>
> ***The scope and study of our paper.***
>
> In our paper, we study and solve *the phishing attack localization problem, specifically in the context of email phishing, that goes beyond email phishing detection* where the corresponding models can only predict the label (phishing or benign) of the emails and often lack the intrinsic ability to automatically learn and figure out the most important and phishing-relevant information (i.e., sentences) that trigger the classification of phishing emails.
>
> Solving the phishing attack localization problem requires a method that can not only predict the vulnerability label (phishing or benign) of the emails, but also automatically identify and learn the most important and phishing-relevant information (e.g., key sentences) causing an email as phishing. To the best of our knowledge, our AI2TALE method is one of the very first approaches proposed for solving the email phishing attack localization problem. *In our paper, we comprehensively presented the related baselines capable of solving the problem as well as compared our method with many recent, popular, and state-of-the-art baselines (i.e., intrinsic interpretability methods, a.k.a. self-explanatory models)*.
>
> ***Regarding the use of the transformer model.***
>
> We might consider fine-tuning a pre-trained transformer model to solve the email phishing attack localization problem. However, when fine-tuning it to learn and identify the most important phishing-relevant information (e.g., key sentences) that cause an email to be classified as phishing, *there is a need for the ground truth of phishing-related information*. As mentioned in Section 3.1 of our paper, in almost all publicly available phishing-relevant data (e.g., emails), there are only labels indicating whether the data is phishing or benign. We almost do not have the ground truth of phishing information (i.e., the information truly causes the data to be classified as phishing). *In the phishing attack localization problem, we name this context as a weakly supervised setting where during the training process, we only use the data’s phishing or benign label while not requiring the ground truth of phishing information of the data.*
>
> In our study, we view each email as a sequence of sentences and aim to identify the most important sentence contributing to the email's vulnerability label (phishing or benign). Through our observations, we found that selecting information at the sentence level provides a more complete and clearer message than selecting at the word level.
>
> We also considered using a pre-trained transformer model (i.e., the encoder part) for word embeddings, which are then used to embed corresponding sentences into vectors. However, we decided not to pursue this option for several reasons. **Firstly**, to facilitate solving the problem effectively, the word and sentence embeddings should be a part of a model that will be updated during the training process. A transformer-based model operates at the token level (e.g., subword level), where the entire input is a sequence of tokens, and the length of the input sequence is also limited. Many common transformer-based models (e.g., BERT, RoBERTa, and T5) typically have an input limit of 512 tokens. Other transformer-based models can handle longer sequences, but they require more computational resources and may involve specialized architectures, such as sparse attention mechanisms or techniques like sliding windows, to efficiently process long inputs. **Secondly**, after obtaining token embeddings from the transformer model, a solution is needed to gain corresponding embedded sentences, which requires appropriate input data organization and computational process (e.g., aligning tokens with their corresponding embeddings, handling padding and truncation, combining token embeddings into a sentence embedding). **Thirdly**, while a pre-trained transformer can perform well on general tasks, fine-tuning it to capture domain-specific nuances may require significant time and appropriate modifications. *Ultimately, we opted for a well-recognized, effective, and lightweight word and sentence embedding solution, as described in our paper.*
>
> ***About using the attention mechanism as mentioned by Reviewer 8txw.***
>
> Compared to the attention mechanism, using information theory, specifically through mutual information, which is grounded in a solid theoretical foundation and widely recognized in explainable AI, provides a more robust and interpretable solution, as demonstrated in some of our baselines.
>
> Regarding the paper that uses the attention mechanism mentioned by the reviewer. In our responses, we have adapted this method to address the email phishing attack localization problem. Both *the experimental results and the stability* of our approach demonstrate its advancements and robustness compared to this method.

---

### Meta-Review · Area_Chair_83Lx · 2024-12-20

**Metareview:**

This work studied the task of phishing email attack localization, and proposed a method based information theory, which cannot only provide prediction, but also the identification of phishing sentence.

It received 3 detailed reviews, with very diverse scores 388.
The strengths mentioned by reviews include: good writing and clear organization, importance of the explainable phishing location task, the technical novelty by using information theory, good experimental results.

Meanwhile, there are also some important concerns, mainly including:
1. The novelty of the studied task.
2. Comparison with attention based method.
3. Using the transformer model.

There are several rounds of discussions between authors and reviewers. I carefully read the manuscript and all discussions. My judgements of about concerns as follows:
1. It is indeed that there are several explainable methods for phishing localization, no matter post-explanation or intrinsic explanation. The claim that "our method is one of the very first approaches proposed for solving the phishing attack localization problem aiming to tackle and improve the explainability" is inappropriate. It is required to remove this claim from the manuscript, and clearly give the credit to existing works.
2. The explanation in the rebuttal that the proposed method is better approach than attention based method is not very convincing as no solid analysis, though the authors added some results to show the better performance of the proposed method. If without very rigorous analysis, it is inappropriate to make such as claim in the manuscript.
3. The authors refused to adopt transformer in the proposed method, with several reasons. But these reasons are unconvincing to both the reviewer and me. Transformer is suitable for handling text based tasks. It is better to add such experiments in the manuscript.

Besides, the authors made efforts to address several other concerns.
Overall, I think the strengths somewhat outweigh its limitations, leading to the recommendation of accept. But, the suggested revisions mentioned above are mandatory in the final version.

**Additional Comments On Reviewer Discussion:**

The rebuttal and discussions, as well as their influences in the decision, have been summarized in the above metareview.

---

### Decision · Program_Chairs · 2025-01-22

Accept (Poster)